# Contrastive learning with Adversarial Perturbations for Conditional Text Generation

**Seanie Lee**[1*], **Dong Bok Lee**[1*], **Sung Ju Hwang**[1,2]
KAIST[1], AITRICS[2], South Korea
{lsnfamily02, markhi, sjhwang82}@kaist.ac.kr

## Abstract

Recently, sequence-to-sequence (seq2seq) models with the Transformer architecture have achieved remarkable performance on various conditional text generation tasks, such as machine translation. However, most of them are trained with *teacher forcing* with the ground truth label given at each time step, without being exposed to incorrectly generated tokens during training, which hurts its generalization to unseen inputs, that is known as the "exposure bias" problem. In this work, we propose to mitigate the conditional text generation problem by contrasting positive pairs with negative pairs, such that the model is exposed to various valid or incorrect perturbations of the inputs, for improved generalization. However, training the model with naïve contrastive learning framework using random non-target sequences as negative examples is suboptimal, since they are easily distinguishable from the correct output, especially so with models pretrained with large text corpora. Also, generating positive examples requires domain-specific augmentation heuristics which may not generalize over diverse domains. To tackle this problem, we propose a principled method to generate positive and negative samples for contrastive learning of seq2seq models. Specifically, we generate negative examples by adding small perturbations to the input sequence to minimize its conditional likelihood, and positive examples by adding large perturbations while enforcing it to have a high conditional likelihood. Such "hard" positive and negative pairs generated using our method guides the model to better distinguish correct outputs from incorrect ones. We empirically show that our proposed method significantly improves the generalization of the seq2seq on three text generation tasks — machine translation, text summarization, and question generation.

## 1 Introduction

The sequence-to-sequence (seq2seq) models (Sutskever et al., 2014), which learn to map an arbitrary-length input sequence to another arbitrary-length output sequence, have successfully tackled a wide range of language generation tasks. Early seq2seq models have used recurrent neural networks to encode and decode sequences, leveraging attention mechanism (Bahdanau et al., 2015) that allows the decoder to attend to a specific token in the input sequence to capture long-term dependencies between the source and target sequences. Recently, the Transformer (Vaswani et al., 2017), which is an all-attention model that effectively captures long-term relationships between tokens in the input sequence as well as across input and output sequences, has become the de facto standard for most of the text generation tasks due to its impressive performance. Moreover, Transformer-based language models trained on large text corpora (Dong et al., 2019; Raffel et al., 2020; Lewis et al., 2020) have shown to significantly improve the model performance on text generation tasks.

However, a crucial limitation of seq2seq models is that they are mostly trained only with *teacher forcing*, where ground truth is provided at each time step and thus never exposed to incorrectly generated tokens during training (Fig. 1-(a)), which hurts its generalization. This problem is known as the "exposure bias" problem (Ranzato et al., 2016) and often results in the generation of low-quality texts on unseen inputs. Several prior works tackle the problem, such as using reinforcement learning (RL) to maximize non-differentiable reward (Bahdanau et al., 2017; Paulus et al., 2018).

---

[*]Equal contribution

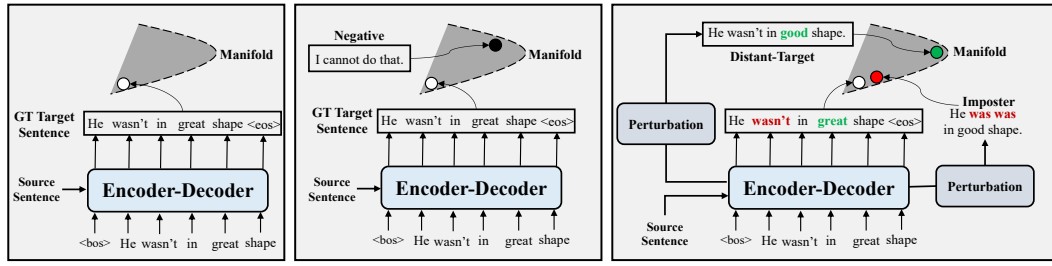

(a) Teacher Forcing    (b) Randomly Sampled Negative Example    (c) Imposter / Distant-Target Generation with perturbation

Figure 1: **Concept.** (a) Training seq2seq with *teacher forcing*. (b) Naïve contrastive learning with randomly sampled negative examples. (c) Our method, CLAPS, which generates hard negative and positive examples.

Another approach is to use RL or gumbel softmax (Jang et al., 2017) to match the distribution of generated sentences to that of the ground truth, in which case the reward is the discriminator output from a Generative Adversarial Network (GAN) (Zhang et al., 2018; 2017; Yu et al., 2017). Although the aforementioned approaches improve the performance of the seq2seq models on text generation tasks, they either require a vast amount of effort in tuning hyperparameters or stabilize training.

In this work, we propose to mitigate the exposure bias problem with a simple yet effective approach, in which we *contrast* a positive pair of input and output sequence to negative pairs, to expose the model to various valid or incorrect sentences. Naïvely, we can construct negative pairs by simply using random non-target sequences from the batch (Chen et al., 2020). However, such a naïve construction yields meaningless negative examples that are already well-discriminated in the embedding space (Fig. 1-(b)), which we highlight as the reason why existing methods (Chen et al., 2020) require large batch size. This is clearly shown in Fig. 2, where a large portion of positive-negative pairs can be easily discriminated without any training, which gets worse as the batch size decreases as it will reduce the chance

Figure 2: Accuracy of classifying a positive pair from negative pairs varying batch size **without any training.**

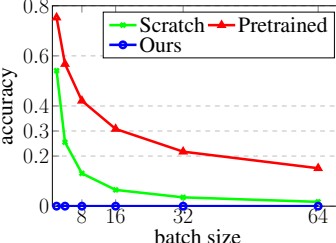

to have meaningfully difficult examples in the batch. Moreover, discriminating positive and naïve negative pairs becomes even more easier for models pretrained on large text corpora.

To resolve this issue, we propose principled approaches to automatically generate negative and positive pairs for constrastive learning, which we refer to as *Contrastive Learning with Adversarial Perturbation for Seq2seq learning* (CLAPS). Specifically, we generate a negative example by adding a small perturbation to the hidden representation of the target sequence, such that its conditional likelihood is minimized (Denoted as the red circle in Fig. 1-(c)). Conversely, we construct an additional positive example (Denoted as green circle in Fig. 1-(c)) by adding a large amount of perturbation to the hidden representation of target sequence such that the perturbed sample is far away from the source sequence in the embedding space, while enforcing it to have high conditional likelihood by minimizing Kullback-Leibler (KL) divergence between the original conditional distribution and perturbed conditional distribution. This will yield a negative example that is very close to the original representation of target sequence in the embedding space but is largely dissimilar in the semantics, while the generated positive example is far away from the original input sequence but has the same semantic as the target sequence. This will generate difficult examples that the model fails to correctly discriminate (Fig. 1-(c), Fig.2), helping it learn with more meaningful pairs.

To verify the efficacy of our method, we empirically show that it significantly improves the performance of seq2seq model on three conditional text generation tasks, namely machine translation, text summarization and question generation. Our contribution in this work is threefold:

- To mitigate the exposure bias problem, we propose a contrastive learning framework for conditional sequence generation, which contrasts a positive pair of source and target sentence to negative pairs in the latent embedding space, to expose the model to various valid or incorrect outputs.

- To tackle the ineffectiveness of conventional approach for constructing negative and positive examples for contrastive learning, we propose a principled method to automatically generate negative and positive pairs, that are more difficult and allows to learn more meaningful representations.

- We show that our proposed method, CLAPS, significantly improves the performance of seq2seq model on three different tasks: machine translation, text summarization, and question generation.

## 2 RELATED WORK

**Exposure Bias** There are several prior works to tackle the exposure bias (Ranzato et al., 2016). Bengio et al. (2015) introduce scheduled sampling where the model is initially guided with the true previous tokens but uses the tokens generated by the seq2seq model as the conditional input for the next token, as training goes on. Paulus et al. (2018); Bahdanau et al. (2017) leverage RL to maximize non-differentiable rewards, so it enables to penalize the model for incorrectly generated sentences. Another works (Zhang et al., 2017; 2018; Yu et al., 2017) train GANs to match the distribution of generated sequences to that of ground truth. Since sampling tokens from the generator is not differentiable, they resort RL or gumbel-softmax to train the networks in end-to-end fashion. However, they require either a large amount of effort to tune hyperparameters or stabilize training. However, Choshen et al. (2020) show that RL for machine translation does not optimize the expected reward and the performance gain is attributed to the unrelated effects such as increasing the peakiness of the output distribution. Moreover, (Caccia et al., 2019) show that by tuning the temperature parameter, the language models trained with MLE can be tuned to outperform GAN-based text generation models.

**Adversarial Perturbation** Many existing works, such as (Madry et al., 2018), address the robustness of neural networks to adversarial examples, which are generated by applying a small perturbations to the input samples. While adversarial robustness has been mostly explored in image domains, Miyato et al. (2017) adopted adversarial training to text domains. However instead of targeting robustness to perturbed samples, they utilize the adversarial examples as augmented data, and enforce consistency across the predictions across original unlabeled example and its perturbation, for semi-supervised learning. Recently Zhu et al. (2019) and Jiang et al. (2020) leverage adversarial training to induce the smoothness of text classifiers, to prevent overfitting to training samples. While they are relevant to ours, these methods do not have the notion of positive and negative examples as they do not consider contrastive learning, and only target text classification. Moreover, they are computationally prohibitive since they use PGD for adversarial training, which requires iterative optimization for each individual sample. Recently, Aghajanyan et al. (2020) propose a simpler yet effective method based on Gaussian noise perturbation to regularize neural networks without expensive PGD steps, which is shown to outperform methods from Zhu et al. (2019) and Jiang et al. (2020). Although our work is similar to these prior works in that we add perturbations to the text embeddings, note that we used the adversarially-generated samples as negative examples of our contrastive learning framework rather than trying to learn the model to be robust to them.

**Contrastive Learning** Contrastive learning has been widely used. It is to learn a representation by contrasting positive pairs and negative pairs. Chopra et al. (2005); Weinberger & Saul (2009); Schroff et al. (2015) leverage a triplet loss to separate positive examples from negative examples in metric learning. Chen et al. (2020) show that contrastive learning can boost the performance of self-supervised and semi-supervised learning in computer vison tasks. In natural language processing (NLP), contrastive learning has been widely used. In Word2Vec (Mikolov et al., 2013), neighbouring words are predicted from context with noise-contrastive estimation (Gutmann & Hyvärinen, 2012). Beyond word representation, Logeswaran & Lee (2018) sample two contiguous sentences for positive pairs and the sentences from other document as negative pairs. They constrast positive and negative pairs to learn sentence representation. Moreover, contrastive learning has been investigated in various NLP tasks — language modeling (Huang et al., 2018), unsupervised word alignment (Liu & Sun, 2015), caption generation (Mao et al., 2016; Vedantam et al., 2017), and machine translation (Yang et al., 2019).

## 3 METHOD

### 3.1 BACKGROUND: CONDITIONAL TEXT GENERATION

The goal of conditional text generation with a seq2seq model is to generate an output text sequence $\mathbf{y}^{(i)} = (y_1^{(i)}, \ldots, y_T^{(i)})$ with length $T$ conditioned on the input text sequence $\mathbf{x}^{(i)} = (x_1^{(i)}, \ldots, x_L^{(i)})$

with length $L$. A typical approach to the conditional text generation is to leverage the encoder-decoder architecture to parameterize the conditional distribution. We maximize the conditional log likelihood $\log p_\theta(\mathbf{y}|\mathbf{x})$ for a given $N$ observations $\{(\mathbf{x}^{(i)}, \mathbf{y}^{(i)})\}_{i=1}^N$ as follows:

$$
\begin{aligned}
\mathcal{L}_{MLE}(\theta) &= \sum_{i=1}^N \log p_\theta(\mathbf{y}^{(i)}|\mathbf{x}^{(i)}) \\
p_\theta(y_1^{(i)}, \ldots, y_T^{(i)}|\mathbf{x}^{(i)}) &= \prod_{t=1}^T p_\theta(y_t^{(i)}|\mathbf{y}_{<t}^{(i)}, \mathbf{x}^{(i)}) \\
p_\theta(y_t^{(i)}|\mathbf{y}_{<t}^{(i)}, \mathbf{x}^{(i)}) &= \mathrm{softmax}(\mathbf{W}\mathbf{h}_t^{(i)} + \mathbf{b}) \\
\mathbf{h}_t^{(i)} &= g(y_{t-1}^{(i)}, \mathbf{M}^{(i)}; \theta), \ \ \mathbf{M}^{(i)} = f(\mathbf{x}^{(i)}; \theta)
\end{aligned}
\tag{1}
$$

where $f, g$ denote the encoder and the decoder respectively and $\mathbf{M}^{(i)} = [\mathbf{m}_1^{(i)} \cdots \mathbf{m}_L^{(i)}] \in \mathbb{R}^{d \times L}$ is the concatenation of the hidden representations of the source tokens $\mathbf{x}^{(i)}$.

## 3.2 Contrastive Learning with Adversarial Perturbations for Seq2Seq

Since most of the seq2seq models are trained with teacher forcing where the ground truth tokens are provided to maximize Eq. 1, they are never exposed to incorrectly generated tokens during training, which is known as the "expousre bias" problem. In order to tackle the problem, we propose a contrastive learning framework to expose the model to various valid or incorrect output sequences for a given input sentence. Following the contrastive learning framework (Chen et al., 2020), we can train the model to learn the representations of the ground truth sentence by contrasting the positive pairs with the negative pairs, where we select the negative pairs as a random non-target output sequence from the same batch. As shown in Fig. 3-(a), we project the source and target text sequences onto the latent embedding space. Then we maximize the similarity between the pair of source and target sequence, while minimizing the similarity between the negative pairs as follows:

$$
\begin{aligned}
\mathcal{L}_{cont}(\theta) &= \sum_{i=1}^N \log \frac{\exp(\mathrm{sim}(\mathbf{z}_{\mathbf{x}}^{(i)}, \mathbf{z}_{\mathbf{y}}^{(i)})/\tau)}{\sum_{\mathbf{z}_{\mathbf{y}}^{(j)} \in S} \exp(\mathrm{sim}(\mathbf{z}_{\mathbf{x}}^{(i)}, \mathbf{z}_{\mathbf{y}}^{(j)})/\tau)} \\
\mathbf{z}_{\mathbf{x}}^{(i)} &= \xi(\mathbf{M}^{(i)}; \theta), \ \mathbf{z}_{\mathbf{y}}^{(i)} = \xi(\mathbf{H}^{(i)}; \theta) \\
\xi([\mathbf{v}_1 \cdots \mathbf{v}_T]; \theta) &:= \mathrm{AvgPool}([\mathbf{u}_1 \cdots \mathbf{u}_T]), \text{ where } \mathbf{u}_t = \mathrm{ReLU}(\mathbf{W}^{(1)}\mathbf{v}_t + \mathbf{b}^{(1)})
\end{aligned}
\tag{2}
$$

where $\xi$ denotes the composition of affine transformation with the ReLU (Nair & Hinton, 2010) and average pooling to compute the fixed sized representation of a sentence $\mathbf{z} \in \mathbb{R}^d$, $\mathbf{H}^{(i)} = [\mathbf{h}_1^{(i)} \cdots \mathbf{h}_T^{(i)}] \in \mathbb{R}^{d \times T}$ is a concatenation of the decoder hidden states of the target sentence $\mathbf{y}^{(i)}$ across all the time steps. Furthermore, $S = \{\mathbf{z}_{\mathbf{y}}^{(j)} : j \neq i\}$ is a set of hidden representations of target sentences (the objects other than circles in Fig. 3-(a)) that are randomly sampled and not paired with the source sentence $\mathbf{x}^{(i)}$, and $\mathrm{sim}(\cdot, \cdot)$ is a cosine similarity function.

However, training the model with naïve contrastive learning framework using random non-target sequences as negative examples is highly suboptimal, as described in the introduction and shown in Fig. 1. Many of such naïve negative examples are often located far away from the positive examples in the embedding space from the beginning, when using the pretrained language model. Therefore, simply using the examples from the same batch as done in Chen et al. (2020) will result in trivial negative examples and require very large batch size to enable sampling meaningful negative pairs within the same batch. Moreover, generating positive examples for text sequences is not a trivial problem either since for text domains, we do not have a well-defined set of augmentation methods that preserves the input semantics, unlike with the image domains. To tackle such difficulties, we propose a principled method to automatically construct the adversarial negative and positive examples, such that the samples are difficult for the model to classify correctly. These adversarial positive/negative pairs can guide the model to learn a more accurate representation of the target text sequence, by identifying which features make the output positive or negative (See Fig. 1-(c)).

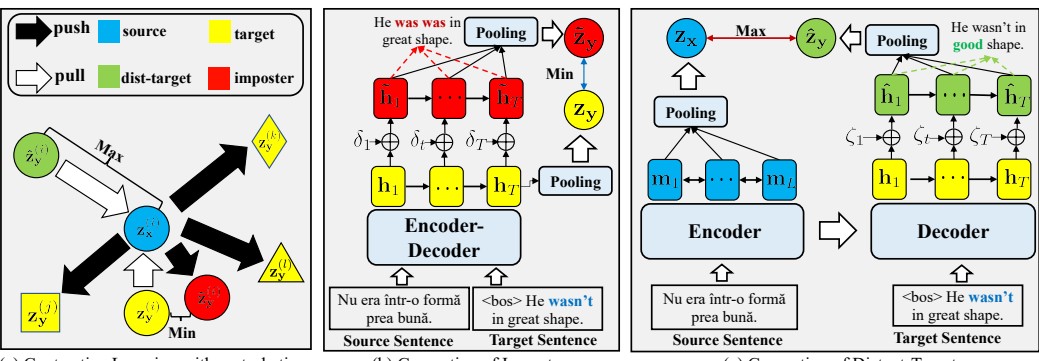

Figure 3: **Generation of imposters and distant-targets with perturbation.** (a) We add small perturbation $\delta_t$ to $\mathbf{h}_t$ for $\tilde{\mathbf{z}}_\mathbf{y}$ so that its conditional likelihood is minimized to generate an invalid sentence. (b) We add large perturbation $\zeta_t$ to $\mathbf{h}_t$ for $\hat{\mathbf{z}}_\mathbf{y}$ by maximizing the distance from $\mathbf{z}_\mathbf{x}$, the representation of source sentence but enforcing its likelihood high to preserve the original semantics.

## 3.3 GENERATION OF IMPOSTERS

As shown in Fig. 3-(b), to generate a negative example, we add a small perturbation $\boldsymbol{\delta}^{(i)} = [\delta_1^{(i)} \cdots \delta_T^{(i)}] \in \mathbb{R}^{d \times T}$ to the $\mathbf{H}^{(i)}$, which is the hidden representation of target sequence $\mathbf{y}^{(i)}$, such that its conditional likelihood is minimized as follows:

$$\tilde{\mathbf{H}}^{(i)} = \mathbf{H}^{(i)} + \boldsymbol{\delta}^{(i)} \text{ where } \boldsymbol{\delta}^{(i)} = \underset{\boldsymbol{\delta}, ||\boldsymbol{\delta}||_2 \leq \epsilon}{\arg \min} \log p_\theta(\mathbf{y}^{(i)}|\mathbf{x}^{(i)}; \mathbf{H}^{(i)} + \boldsymbol{\delta})$$

$$p_\theta(\mathbf{y}^{(i)}|\mathbf{x}^{(i)}; \mathbf{H}^{(i)} + \boldsymbol{\delta}) = \prod_{t=1}^{T} p_\theta(y_t^{(i)}|\mathbf{y}_{<t}^{(i)}, \mathbf{x}^{(i)}; \mathbf{h}_t^{(i)} + \delta_t) \tag{3}$$

$$p_\theta(y_t^{(i)}|\mathbf{y}_{<t}^{(i)}, \mathbf{x}^{(i)}; \mathbf{h}_t^{(i)} + \delta_t) = \text{softmax}\{\mathbf{W}(\mathbf{h}_t^{(i)} + \delta_t) + \mathbf{b}\}, \text{ where } \delta_t \in \mathbb{R}^d$$

The exact minimization of the conditional log likelihood with respect to $\boldsymbol{\delta}$ is intractable for deep neural networks. Following Goodfellow et al. (2015), we approximate it by linearizing $\log p_\theta(\mathbf{y}^{(i)}|\mathbf{x}^{(i)})$ around $\mathbf{H}^{(i)}$ as follows:

$$\tilde{\mathbf{H}}^{(i)} = \mathbf{H}^{(i)} - \epsilon \frac{\boldsymbol{g}}{||\boldsymbol{g}||_2}, \text{ where } \boldsymbol{g} = \nabla_{\mathbf{H}^{(i)}} \log p_\theta(\mathbf{y}^{(i)}|\mathbf{x}^{(i)}) \tag{4}$$

We add small perturbation to the hidden representation of each token of target sentence $\mathbf{y}^{(i)}$ such that its conditional likelihood is minimized. Thus, the perturbed $\tilde{\mathbf{H}}^{(i)}$, which we call an *imposter* (inspired by Weinberger & Saul (2009)), is semantically very dissimilar to $\mathbf{y}^{(i)}$, but very close to the hidden representation $\mathbf{H}^{(i)}$ in the embedding space (Fig. 3-(a)). This will make it non-trivial for the sequence-to-sequence model to distinguish it from the representation of true target sequence $\mathbf{y}^{(i)}$. Please note while adversarial perturbations are generated similarly as in Miyato et al. (2017), we use them in a completely different way. While they train the model to be invariant to adversarial samples within the $\epsilon$-ball, we push them far away from the source sentence while pulling the ground truth target sentence to the input sentence. In other words, we use the perturbed representation as an additional negative sample for contrastive learning as follows:

$$\mathcal{L}_{cont-neg}(\theta) = \sum_{i=1}^{N} \log \frac{\exp(\text{sim}(\mathbf{z}_\mathbf{x}^{(i)}, \mathbf{z}_\mathbf{y}^{(i)})/\tau)}{\sum_{\mathbf{z}_\mathbf{y}^{(k)} \in S \cup \{\tilde{\mathbf{z}}_\mathbf{y}^{(i)}\}} \exp(\text{sim}(\mathbf{z}_\mathbf{x}^{(i)}, \mathbf{z}_\mathbf{y}^{(k)})/\tau)}, \text{ where } \tilde{\mathbf{z}}_\mathbf{y}^{(i)} = \xi(\tilde{\mathbf{H}}^{(i)}; \theta) \tag{5}$$

Alternatively, we can generate an imposter by perturbing the hidden representation of target sentence $\mathbf{y}$ so that its conditional likelihood is minimized but very close to the source sentence $\mathbf{x}$ in the embedding space. However, we empirically find that such a variation yields less performance gain.

## 3.4 GENERATION OF DISTANT-TARGETS

Moreover, as shown in Fig. 3-(c), we construct an additional positive pair of source sequence $\mathbf{x}^{(i)}$ by adding large perturbation $\zeta^{(i)} = [\zeta_1^{(i)} \cdots \zeta_T^{(i)}] \in \mathbb{R}^{d \times T}$ to $\mathbf{H}^{(i)}$ the hidden state of target sequence $\mathbf{y}^{(i)}$, such that cosine similarity from $\mathbf{z}_\mathbf{x}^{(i)}$ is minimized, but the conditional likelihood is

enforced to remain high. However, the exact computation of $\zeta^{(i)}$ with such constraints is intractable. We approximate it with the following two separate stages. First, we add perturbation to $\mathbf{H}^{(i)}$ such that it minimizes the contrastive learning objective $\mathcal{L}_{cont}(\theta)$ as shown in Eq. 6. Then we add another perturbation to minimize the KL divergence between perturbed conditional distribution $p_\theta(\hat{y}_t^{(i)}|\hat{\mathbf{y}}_{<t}^{(i)}, \mathbf{x}^{(i)})$ and the original conditional distribution $p_\theta(y_t^{(i)}|\mathbf{y}_{<t}^{(i)}, \mathbf{x}^{(i)})$ as shown in Eq. 7, where $\overline{\mathbf{H}} = [\overline{\mathbf{h}}_1 \cdots \overline{\mathbf{h}}_T] \in \mathbb{R}^{d \times T}$, $\hat{\mathbf{H}} = [\hat{\mathbf{h}}_1 \cdots \hat{\mathbf{h}}_T] \in \mathbb{R}^{d \times T}$, and $\eta \in \mathbb{R}$. Note that $\theta^*$ denotes the copied of the model parameter $\theta$ and is considered to be constant to prevent it from being updated through back-propagation.

$$\overline{\mathbf{H}}^{(i)} = \mathbf{H}^{(i)} - \eta \frac{\mathbf{g}}{||\mathbf{g}||_2} \text{ where } \mathbf{g} = \nabla_{\mathbf{H}^{(i)}} \mathcal{L}_{cont}(\theta) \tag{6}$$

$$p_\theta(\hat{y}_t^{(i)}|\hat{\mathbf{y}}_{<t}^{(i)}, \mathbf{x}^{(i)}) = \text{softmax}(\mathbf{W}\overline{\mathbf{h}}_t^{(i)} + \mathbf{b})$$

$$\mathcal{L}_{KL}(\theta) = \sum_{i=1}^{N} \sum_{t=1}^{T} D_{KL}(p_{\theta^*}(y_t^{(i)}|\mathbf{y}_{<t}^{(i)}, \mathbf{x}^{(i)})||p_\theta(\hat{y}_t^{(i)}|\hat{\mathbf{y}}_{<t}^{(i)}, \mathbf{x}^{(i)})) \tag{7}$$

$$\hat{\mathbf{H}}^{(i)} = \overline{\mathbf{H}}^{(i)} - \eta \frac{\mathbf{f}}{||\mathbf{f}||_2}, \text{ where } \mathbf{f} = \nabla_{\overline{\mathbf{H}}_1^{(i)}} \mathcal{L}_{KL}(\theta)$$

We consider the perturbed hidden state $\hat{\mathbf{H}}^{(i)}$ as an additional positive example for source sequence $\mathbf{x}^{(i)}$, which we refer to as a *distant-target*. We can use a distant-target to augment contrastive learning and minimize $\mathcal{L}_{KL}(\theta)$ as follows:

$$\mathcal{L}_{cont-pos}(\theta) = \sum_{i=1}^{N} \log \frac{\exp(\text{sim}(\mathbf{z}_\mathbf{x}^{(i)}, \hat{\mathbf{z}}_\mathbf{y}^{(i)})/\tau)}{\sum_{\mathbf{z}_\mathbf{y}^{(k)} \in S \cup \{\tilde{\mathbf{z}}_\mathbf{y}^{(i)}\}} \exp(\text{sim}(\mathbf{z}_\mathbf{x}^{(i)}, \mathbf{z}_\mathbf{y}^{(k)})/\tau)}, \text{ where } \hat{\mathbf{z}}_\mathbf{y}^{(i)} = \xi(\hat{\mathbf{H}}^{(i)}; \theta) \tag{8}$$

**CLAPS objective** Incorporating the loss on the imposter and the distant target introduced above, we estimate the parameters of the seq2seq model $\theta$ by maximizing the following objective, where $\alpha, \beta$ are hyperparameters which control the importance of contrastive learning and KL divergence:

$$\max_\theta \mathcal{L}_{MLE}(\theta) - \alpha \mathcal{L}_{KL}(\theta) + \beta \{\mathcal{L}_{cont-neg}(\theta) + \mathcal{L}_{cont-pos}(\theta)\} \tag{9}$$

For all the experiments, we set $\alpha$ and $\beta$ as 1, which we search through cross-validation. Note that after training is done, we remove the pooling layer $\xi$ and generate text with the decoder $g$, given an input encoded with the encoder $f$.

## 4 EXPERIMENT

We validate our method on benchmark datasets on three conditional text generation tasks.

### 4.1 TASKS

**Machine Translation (MT)** For machine translation, we use WMT16 Romanian-English parallel corpus (WMT'16 RO-EN) to train the model. We tokenize the pairs of source and target sequences with the same tokenizer as Raffel et al. (2020). We finetune the pretrained T5-small model for 20 epochs with the batch size of 128 and Adafactor (Shazeer & Stern, 2018). For contrastive learning, we set the norm of perturbation, $\eta$ and $\epsilon$ as 3.0.

**Text Summarization (Sum.)** For text summarization, we use XSum dataset (Narayan et al., 2018) of which summaries are highly abstractive, thus extractive summarization models under-perform abstractive models. We follow the most of the experimental settings for machine translation as described above, except that we set the norm of perturbation, $\eta$ and $\epsilon$ as 1.0 and 1.0, respectively.

**Question Generation (QG)** For question generation, we aim to generate a question from a given answer and paragraph, i.e., we model conditional distribution $p_\theta(\mathbf{y}|\mathbf{x}, \mathbf{a})$ where $\mathbf{x}, \mathbf{y}, \mathbf{a}$ denote a paragraph, question and answer, respectively. We concatenate the answer and paragraph with special tokens to generate the question conditioned on both of the answer and paragraph. As the previous experimental settings, we finetune T5-small model on SQuAD dataset (Rajpurkar et al., 2016) for 20 epochs with batch size 128 and set the norm of perturbation, $\eta$ as 3.0 and $\epsilon$ as 1.0. Since the test set of SQuAD is only accessible via leader board, we randomly split the validation set into a validation set and a test set.

| Method | Aug. | BLEU-1 | BLEU-2 | BLEU-3 | BLEU-4 | BLEU | F1/EM |
|---|---|---|---|---|---|---|---|
| **Question Generation** - SQuAD | | | | | | | |
| Harvesting-QG | - | - | - | 20.90 | 15.16 | - | 66.05/54.62 |
| T5-MLE | - | 41.26 | 30.30 | 23.38 | 18.54 | 21.00 | 67.64/55.91 |
| $\alpha$-T5-MLE ($\alpha = 0.7$) | - | 40.82 | 29.79 | 22.84 | 17.99 | 20.50 | 68.04/56.30 |
| $\alpha$-T5-MLE ($\alpha = 2.0$) | - | 37.35 | 27.20 | 20.79 | 16.36 | 18.41 | 65.74/54.76 |
| T5-SSMBA | Pos. | 41.67 | 30.59 | 23.53 | 18.57 | 21.07 | 68.47/56.37 |
| T5-WordDropout Contrastive | Neg. | 41.37 | 30.50 | 23.58 | 18.71 | 21.19 | 68.16/56.41 |
| R3F | - | 41.00 | 30.15 | 23.26 | 18.44 | 20.97 | 65.84/54.10 |
| T5-MLE-contrastive | - | 41.23 | 30.28 | 23.33 | 18.45 | 20.91 | 67.32/55.25 |
| **T5-CLAPS w/o negative** | Pos. | 41.87 | 30.93 | 23.90 | 18.92 | 21.38 | - |
| **T5-CLAPS w/o positive** | Neg. | 41.65 | 30.69 | 23.71 | 18.81 | 21.25 | 68.26/56.41 |
| **T5-CLAPS** | Pos.+Neg. | **42.33** | **31.29** | **24.22** | **19.19** | **21.55** | **69.01/57.06** |
| ERNIE-GEN (Xiao et al., 2020) | - | - | - | - | **26.95** | - | - |
| Info-HCVAE (Lee et al., 2020) | - | - | - | - | - | - | **81.51/71.18** |
| **Machine Translation** - WMT'16 RO-EN | | | | | | | |
| Transformer | - | 50.36 | 37.18 | 28.42 | 22.21 | 26.17 | |
| Scratch-T5-MLE | - | 51.62 | 37.22 | 27.26 | 21.13 | 25.34 | |
| Scratch-CLAPS | Pos.+Neg. | 53.42 | 39.57 | 30.24 | 23.59 | 27.61 | |
| T5-MLE | - | 57.76 | 44.45 | 35.12 | 28.21 | 32.43 | |
| $\alpha$-T5-MLE ($\alpha = 0.7$) | - | 57.63 | 44.23 | 33.84 | 27.90 | 32.14 | |
| $\alpha$-T5-MLE ($\alpha = 2.0$) | - | 56.03 | 42.59 | 33.29 | 26.45 | 30.72 | |
| T5-SSMBA | Pos. | 58.23 | 44.87 | 35.50 | 28.48 | 32.81 | |
| T5-WordDropout Contrastive | Neg. | 57.77 | 44.45 | 35.12 | 28.21 | 32.44 | |
| R3F | - | 58.07 | 44.86 | 35.57 | 28.66 | 32.99 | |
| T5-MLE-contrastive | - | 57.64 | 44.12 | 34.74 | 27.79 | 32.03 | |
| **T5-CLAPS w/o negative** | Pos. | 58.81 | 45.52 | 36.20 | 29.23 | 33.50 | |
| **T5-CLAPS w/o positive** | Neg. | 57.90 | 44.60 | 35.27 | 28.34 | 32.55 | |
| **T5-CLAPS** | Pos.+Neg. | **58.98** | **45.72** | **36.39** | **29.41** | **33.96** | |
| Conneau & Lample (2019) | - | - | - | - | - | **38.5** | |

Table 1: BLEU scores on WMT'16 RO-EN and SQuAD for machine translation and question generation. EM/F1 scores with BERT-base QA model for question generation.

## 4.2 EXPERIMENTAL SETUPS

**Implementation Details** For the encoder $f$, and decoder $g$, we use T5-small model, which is based on transformer with the hidden dimension, $d = 512$. We set the temperature, $\tau$ as 0.1 for all the experiments. At test time, we use beam search of width 4 to generate the target sequences. **Common Baselines** We compare our method against relevant baselines.

1. **T5-MLE**: A pretrained T5 model fine-tuned to maximize $\mathcal{L}_{MLE}(\theta)$.
2. **Scratch-T5-MLE**: A random initialized Transformer model that has the identical architecture to T5, trained by maximizing $\mathcal{L}_{MLE}(\theta)$.
3. **$\alpha$-T5-MLE**: T5 model trained with MLE, with varying temperature $\alpha$ in the softmax function when decoding the target sentences, as done in Caccia et al. (2019)
4. **T5-SSMBA**: This is the T5 model trained to maximize $\mathcal{L}_{MLE}(\theta)$, with additional examples generated by the technique proposed in Ng et al. (2020). which are generated by corrupting the target sequences and reconstructs them using a masked language model, BERT.
5. **T5-WordDropout Contrastive**: This is a T5 model trained with the contrastive learning framework proposed in Yang et al. (2019), which heuristically generates negative examples by removing the most frequent word from the target sequence. We pretrain T5-small to maximize $\mathcal{L}_{MLE}(\theta)$ and further train the model to assign higher probability to the ground truth target sentence than a negative example with max-margin loss.
6. **R3F**: This is a T5 model that minimizes the negative log likelihood and symmetric KL-divergence between original conditional log likelihood $p_\theta(\mathbf{y}|\mathbf{x})$ and $p_\theta(\mathbf{y}|\tilde{\mathbf{x}})$ to enforce the function to be smooth, where $\tilde{\mathbf{x}} = \text{WordEmbedding}(\mathbf{x}) + \mathbf{z}, \mathbf{z} = (z_1, \ldots, z_L), z_i \overset{i.i.d}{\sim} \mathcal{N}(\mathbf{0}, \text{diag}(\sigma_1, \ldots, \sigma_d))$.
7. **T5-MLE-contrastive**: This is a naive constrastive learning framework with positive/negative pairs, which maximizes the contrastive learning objective from Eq. 2.
8. **T5-CLAPS w/o positive (negative)**: Our proposed model which jointly maximizes the log likelihood and the contrastive learning objective with imposters but does not use any distant-targets or imposters.

Table 2: Rouge and Meteor on Xsum test set for text summarization.

| Method | Aug. | Rouge-1 | Rouge-2 | Rouge-L | METEOR |
|---|---|---|---|---|---|
| **Text Summarization** - XSum | | | | | |
| PTGEN-COVG | - | 28.10 | 8.02 | 21.72 | 12.46 |
| CONVS2S | - | 31.89 | 11.54 | 25.75 | 13.20 |
| Scratch-T5-MLE | - | 31.44 | 11.07 | 25.18 | 13.01 |
| Stcratch-CLAPS | Pos.+Neg. | 33.52 | 12.59 | 26.91 | 14.18 |
| T5-MLE | - | 36.10 | 14.72 | 29.16 | 15.78 |
| $\alpha$-T5-MLE ($\alpha = 0.7$) | - | 36.68 | 15.10 | 29.72 | 15.78 |
| $\alpha$-T5-MLE ($\alpha = 2.0$) | - | 34.18 | 13.53 | 27.35 | 14.51 |
| T5-SSMBA | Pos. | 36.58 | 14.81 | 29.68 | 15.38 |
| T5-WordDropout Contrastive | Neg. | 36.88 | 15.11 | 29.79 | 15.77 |
| R3F | - | 36.96 | 15.12 | 29.76 | 15.68 |
| T5-MLE-contrastive | - | 36.34 | 14.81 | 29.41 | 15.85 |
| **T5-CLAPS w/o negative** | Pos. | 37.49 | 15.31 | 30.42 | 16.36 |
| **T5-CLAPS w/o positive** | Neg. | 37.72 | 15.49 | **30.74** | 16.06 |
| **T5-CLAPS** | Pos.+Neg. | **37.89** | **15.78** | 30.59 | **16.38** |
| PEGASUS (Zhang et al., 2020) | - | **47.21** | **24.56** | **39.25** | - |

9. **T5-CLAPS**: Our full model which jointly maximizes the log likelihood, contrastive learning objective, and KL-divergence as described in the Eq. 9.

10. **Scratch-CLAPS**: Our full model as **T5-CLAPS** but with randomly initialized T5 architecture.

**Task specific baselines** For machine translation, we use the Transformer (Vaswani et al., 2017) which consists of 6 layers of self-attention layer with 8 multi-head attention and 512 dimension, as an additional baseline. For QG, we additionally compare our models against Harvesting-QG (Du & Cardie, 2018), which is a LSTM model with copy mechanism. For text summarization, we use PTGEN-COVG (See et al., 2017) as a baseline, which uses copy mechanism and coverage to handle out of vocabulary word and prevent word repetition, and CONVS2S (Narayan et al., 2018) which uses convolutional networks as the encoder and decoder.

**Evaluation Metric** Following the conventional evaluation metrics, we adopt n-gram BLEU and BLEU (Papineni et al., 2002) for MT and QG. For text summarization, we use Rouge (Lin & Hovy, 2002) and Meteor (Banerjee & Lavie, 2005). As an additional performance measure for question generation, we evaluate a BERT QA model on the SQuAD test set, where the QA model is trained with the questions generated by each QG methods from the contexts and answers of HarvestingQA dataset (Du & Cardie, 2018), and report the F1 and Exact Match (EM).

## 4.3 EXPERIMENTAL RESULTS

**Quantitative Results** We compare our model with the baseline models on WMT'16 RO-En, XSum, SQuAD dataset for machine translation, text summarization and question generation, respectively. Table 1 shows that our proposed method CLAPS significantly outperforms the other baseline, with the performance gain of more than $1\%$ on all tasks according to the BLEU scores. Moreover our proposed method improves the performance of the randomly initialized T5 model (Scratch-CLAPS). For question generation, our proposed method also improves F1/EM as well as BLEU scores. It shows that our proposed model is able to generate semantically valid questions that are beneficial for training the QA model. Note that naively constructing the negative examples for contrastive learning on the both tasks, by randomly shuffling the association of $(\mathbf{x}, \mathbf{y})$ from a given mini-batch, degrades the performance. Increasing the batch size to a large value, using larger memory, may increase its performance as observed in SimCLR (Chen et al., 2020). However, such an approach will be highly sample-inefficient. On the contrary, our model outperforms all the other baseline models on Xsum dataset for text summarization, as shown in Table 2. For summarization, we observe that contrastive learning with imposters alone can improve the performance by a large margin.

**Visualization** To examine our model with proposed contrastive learning framework learns meaningful representation of sentences, we encode a pair of sequences $(\mathbf{x}, \mathbf{y})$ into $\mathbf{M}$, $\mathbf{H}$ with encoder $f$ and decoder $g$. Then, we add perturbations to $\mathbf{H}$ to construct an imposter $\tilde{\mathbf{H}}$ and an additional positive

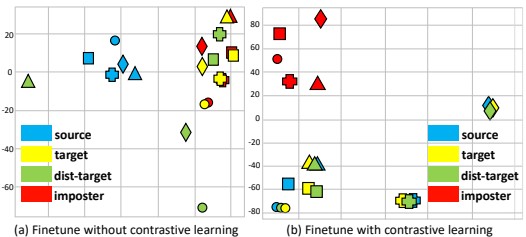

(a) Finetune without contrastive learning  (b) Finetune with contrastive learning

| | |
| --- | --- |
| **(MT)** Lupta lui Hilary a fost mai atractivă. | |
| =>(**GT**): Hillary's **struggle** was more attractive | |
| =>(**Dist.**): Hilary's **fight** was more attractive | |
| =>(**Imp.**): **Thearies'** battle fight has attractive appealing | |

| |
| --- |
| **(QG)** … Von Miller … recording **five** solo tackles, … |
| =>(**GT**): How many solo tackles did Von Miller **make** at Super Bowl 50? |
| =>(**Dist.**): How many solo tackles did Von Miller **record** at Super Bowl 50? |
| =>(**Imp.**): What much tackle **did was** Miller record at Super Bowl 50? |

| |
| --- |
| **(Sum.)** Pieces from the board game … have been found in … China. … |
| =>(**GT**): An ancient board game has been **found** in a Chinese Tomb. |
| =>(**Dist.**): An ancient board game has been **discovered** in a Chinese Tomb. |
| =>(**Imp.**): America's gained vast Africa **most well geographical** countries, 22 |

Figure 4: **Visualization.** (a) Embedding space without contrastive learning. (b) Embedding space with our proposed contrastive learning, CLAPS.

Table 3: Greedy decoding from hidden representation of imposters and distant-targets. The answer span is highlighted for QG.

example $\hat{H}$ as shown in Eq. 3 and 6, 7. We apply average pooling to $M, H, \tilde{H}$, and $\hat{H}$ and project them onto two dimensional space with t-SNE (Maaten & Hinton, 2008). As shown in Fig. 4-(b), the model pushes away the imposter from the embedding of target sequence and pulls the embedding of the distant-targets to the embedding of the source sequence. For the model without contrastive learning, however, the embeddings of both target sequences and distant targets are far away from those of source sequences and the imposters are very close to them as shown in Fig. 4-(a).

**Qualitative Examples** For qualitative analysis, we examine the texts that are represented by the distant-target and imposter from our method, CLAPS. To decode them into output sequences, we apply affine transformation and softmax to $\tilde{H}$ and $\hat{H}$ and select the most likely token at each time step. As shown in Table 3, the distant-target example (**Dist.**), preserves the semantic of the original target sequence (**GT**) with a single word replaced by a synonym (colored in green). However, the imposters (**Imp.**) have completely different semantics, and often are gramatically incorrect (colored in red). This shows that the model are exposed to those various valid or incorrect sentences with our proposed contrastive learning framework with adversarial perturbations.

**Human Evaluation** We further conduct a human evaluation of the 20 summaries and 20 questions generated by our CLAPS and T5-MLE trained for text summarization and QG task. Specifically, 20 human judges perform blind quality assessment of two sentences generated by the two models, that are presented in a random order. For text summarization, **70%** of the human annotators chose the sentences generated by our model as better than the baseline, and for QG, **85%** favored the sentences generated by our model over that of the baseline.

## 5 CONCLUSION

To mitigate the exposure bias problem in sequence-to-sequence learning, we proposed a contrastive learning framework which maximizes the similarity between ground truth input and output sequence, and minimize the similarity between the input and an incorrect output sequence. Moreover, since conventional approach to sample random non-target examples from the batch as negative examples for contrastive learning results in trivial pairs that are well-discriminated from the beginning, we propose a new principled approach to automatically construct "hard" negative and positive examples, where the former is semantically dissimilar but close to the input embedding, and the latter is far from the input embedding but semantically similar. This adversarial learning enables the model to learn both the correct and incorrect variations of the input, and generalize better to unseen inputs. We empirically showed that our method improved the performance of seq2seq model on machine translation, question generation, and text summarization tasks. While we specifically targeted the exposure bias problem with seq2seq models for conditional text generation, our method may be applicable to seq2seq learning for tasks from other domains, such as automatic speech recognition, text-to-speech generation, or video captioning.

**Acknowledgements** This work was supported by Institute of Information & communications Technology Planning & Evaluation (IITP) grant funded by the Korea government (MSIT) (No.2020-0-00153), Samsung Advanced Institute of Technology (SAIT), Samsung Electronics (IO201214-08145-01), Institute of Information & communications Technology Planning & Evaluation (IITP) grant funded by the Korea government (MSIT) (No.2019-0-00075, Artificial Intelligence Graduate School Program (KAIST)), and the Engineering Research Center Program through the National Research Foundation of Korea (NRF) funded by the Korean Government MSIT (NRF-2018R1A5A1059921).

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

Table 4: The statistics and the data source of WMT'16 RO-EN, Xsum, and SQuAD.

| Datasets | Train (#) | Valid (#) | Test (#) | Source |
|---|---|---|---|---|
| WMT'16 RO-EN | 610,320 | 1,999 | 1,999 | Romanian-English Parallel corpus. |
| Xsum | 204,045 | 11,332 | 11,334 | One-sentence summary of BBC news articles. |
| SQuAD | 86,588 | 5,192 | 5,378 | Crowd-sourced questions from Wikipedia paragraph |

## A EXPERIMENTAL DETAILS

**Dataset** For machine translation, text summarization, and question generation, we use WMT'16 RO-EN, Xsum, SQuAD dataset, for each task. The number of train/validation/test set and its source is shown in Table 4. Note that the number of validation and test set for SQuAD is different from the original dataset. Since the original test set is only accessible via the leader board of SQuAD[1], we split the original validation set into our new validation and test set, following the conventions of question generation communities.

**Preprocessing** For machine translation, we download the raw text[2], not the tokenized text, and use the same T5-tokenizer as Raffel et al. (2020) to tokenize both Romanian and English sentences. We limit the input and output length to 128 tokens. For text summarization, we also use the T5-tokenizer as before, and limit the input length to 512 tokens and output length to 128 tokens. For question generation, we set the maximum length of question as 64 tokens and input which is concatenation of answer and context as 384 tokens.

**Implementation** We finetune the pretrained T5-small model provided from the transformers library (Wolf et al., 2019)[3] with Adafactor optimizer. We set the batch size 128 and follow the default setting of Adafactor optimizer to finetune the T5-small models. However, the number of negative examples from the batch is 16 or 32 (total batch size divided by the number of GPUs), because we split the batch into smaller batches and distribute them to each GPU machines. We use 8 GPUs for text summarization, and 4 GPUs for machine translation and question generation. The dimension of hidden state of T5 model, $d$ is 512, so we set the hidden size of $\mathbf{z}$ as the same.

**Evaluation** We use beam search with beam width 4 to generate the target sentences from the source sentences of the test set. Some of the examples are shown in Table 5,6,A. After the generation, we convert the tokens into the raw texts and compare them to the raw text of ground truth target sentences with the automatic evaluation metrics. For n-gram BLEU and Meteor, we use the implementation by Sharma et al. (2017)[4]. For BLEU score, we adopt the implementation by Post (2018)[5].

---

[1]https://rajpurkar.github.io/SQuAD-explorer/
[2]https://s3.amazonaws.com/datasets.huggingface.co/translation/wmt_en_ro.tar.gz
[3]https://github.com/huggingface/transformers
[4]https://github.com/Maluuba/nlg-eval
[5]https://github.com/mjpost/sacrebleu

Table 5: Generated summaries by CLAPS from Xsum dataset.

**Article**: The US military says a strike targeting Taliban in the northern city of
Kunduz may have caused "collateral damage". Offering his "deepest condolences", Mr Obama said
he expected a "full accounting of the facts" and would then make a definitive judgement. . . .

**GT:** President Barack Obama says the US has launched a "full investigation" into air strikes that
killed 19 people at an MSF-run Afghan hospital on Saturday.

**CLAPS:** US President Barack Obama has called for an inquiry into air strikes in Afghanistan that
killed dozens of medical workers.

**Article**: Forecasts were for quarterly growth of between 0.5% and 0.7%. Official statistics also
showed that household consumption expenditure boosted the quarterly growth numbers.
But economist Shane Oliver told the BBC the numbers were "well below potential".
On an annual basis the economy expanded 2.3%, beating expectations for 2.1%.
Economic growth in the March quarter of 2014 was 2.9%. "The March quarter GDP
[gross domestic product] growth was far better than feared just a few days ago,"
said Mr Oliver, who is chief economist with AMP Capital in Sydney.
"However, Australia is still not out of the woods, as annual growth at 2.3% is well below potential,
and a full 0.8% percentage points of the 0.9% growth came from higher inventories and trade."
He said domestic demand remained "very weak with consumer spending
and home construction only just offsetting the ongoing slump in mining investment". ...

**GT**: Australia's economy grew at a better-than-expected 0.9% in the first quarter of 2015,
compared to the previous quarter, boosted by mining together with financial and insurance services.

**CLAPS**: Australia's economy grew faster than expected in the first three months of the year,
according to official figures.

**Article:** After the problems last week, many doubt the system will cope.
Transport for London (TfL) remains confident, although it admits there will be breakdowns.
The trick will be in getting the system back up and running quickly. So here's some friendly advice
for tourists and Olympic visitors to try and make the transport experience as easy as possible.
If anyone thinks of any more please post below.

**GT:** The busiest summer ever looms for London's transport system.

**CLAPS:** London's transport system has been a pretty busy week.

**Article:** The outgoing vice-president spoke during a state dinner and took the opportunity to praise
America's northern neighbour. "The world is going to spend a lot of time looking to you,
Mr Prime Minister", he told the Canadian leader. Mr Biden has been highly critical of
US President-elect Donald Trump. "Vive le Canada because we need you very, very badly,"
he told the dinner guests. He went on to describe the self-doubt that liberal leaders across the world
are currently experiencing after several political defeats. But he praised "genuine leaders" including
German Chancellor Angela Merkel, saying such statesmen and women are in short supply.
Mr Trudeau reportedly became emotional during Mr Biden's remarks when the American
spoke of his late father, former Prime Minister Pierre Trudeau.
"You're a successful father when your children turn out better than you," Mr Biden said. ...

**GT:** US Vice-President Joe Biden told an audience in Ottawa that the world needs "genuine leaders"
such as Canadian Prime Minister Justin Trudeau.

**CLAPS:** Vice-President Joe Biden has praised Canadian Prime Minister Vive le Canada
for his visit to the country.

**Article**: The Swedish giant asked customers who bought
any model of the Mysingso chair to return it for a full refund.
The global recall comes after Ikea received reports from
Finland, Germany, the US, Denmark and Australia
that users had received injuries to their fingers that needed medical treatment.
Ikea's statement said the chair had a "risk of falling or finger entrapment".
It said: "After washing the fabric seat it is possible to re-assemble the chair incorrectly
leading to risks of falls or finger entrapments.
"Ikea has received five incident reports in which a Mysingso beach chair collapsed
during use due to incorrect re-assembly.
All five reports included injuries to fingers and required medical attention.
It added that a full investigation had led to an improved design
"to further mitigate the risks of incorrect re-assembly and injuries"
and the updated chair would be available from next month.
Ikea has more than 300 stores in 27 countries.

**GT:** Ikea is recalling a beach chair sold in the UK after reports
that it can collapse and cause injury.

**CLAPS:** Ikea is recalling a popular beach chair that collapsed during
use because of incorrect re-assemblies.

**Article:** Spending on the NHS should also be paid for
by a dedicated tax marked on every payslip, the former health minister suggested.
Under Mr Lamb's plan, taxes would not be increased as the new levy would be offset
by deductions to income tax or national insurance.
He has warned the NHS faces collapse without an urgent cash injection.
The plans are not yet party policy and will not be put to
this year's conference in Bournemouth. But Mr Lamb, the party's health spokesman,
told party members he was "very interested in the idea of a dedicated NH
S and care contribution - separating it out from the rest of taxation,
clearly identified on your payslip. "And I am really interested in the idea
of the right for local areas to raise additional funds for the NHS
and care if they choose." The Lib Dems say he would like to implement
the ideas across the UK, although, as health and social care are devolved,
it is unclear how this would be enforced.
Mr Lamb - who lost out to Tim Farron in a leadership election in July
- proposes a cross-party commission to explore the ideas.
He intends to consult health bodies and professionals,
patients, trade unions and academics. Ministers have pledged £2bn in this financial year
for the NHS, and an extra £8bn by 2020.
But Mr Lamb told the BBC that this was insufficient and,
having "seen the books" as a minister in the last government,
he feared the NHS could face a funding shortfall of £30bn by 2020.
"The bottom line is with rising demand because of an ageing population we need more investment,"
he said. Mr Lamb also warned that the social care system was "on its knees"
and could collapse without a cash injection of £5bn.
"I've been in the department. I have seen the books and I am deeply concerned.
If we carry on regardless, the system will crash."
Taxpayers are already shown how much they have contributed to the health service
in annual personal tax statements. An attempt to establish a cross-party commission
on social care before the 2010 election - led in part by Mr Lamb - collapsed in acrimony.

**GT**: English councils should be allowed to put up taxes to fund the NHS,
Norman Lamb has told the Lib Dem conference.

**CLAPS**:A new levy on the NHS and social care should be
introduced by the Liberal Democrats, Norman Lamb has said.

**Article**: Yorkshire, Lancashire and Derbyshire have been worst affected,
after 2-5cm fell overnight, with 10cm reported on higher ground.
Passengers waiting to depart Manchester Airport have reported
being stuck on the runway for hours due to a lack of de-icers.
Leeds Bradford Airport suspended all morning flights but has since reopened.
Manchester Airport reported "minor delays to departing aircraft"
- but passengers told the BBC they had been stuck on board outbound flights.
Shirley Hale said her Jet2 flight to Tenerife
had been waiting to depart for over four hours.
"We have been told that there are not enough de-icers at the airport,"
she said. The airport apologised and said de-icing was the responsibility of
airlines and their ground teams. More than 100 schools were closed across East
Lancashire and Oldham, with 80 shut in West Yorkshire.
BBC Weather said Buxton in Derbyshire saw up to 17cm of snow,
the deepest measured on Friday. The avalanche risk in the Peak District was
currently extremely high, Buxton Mountain Rescue Team said.
Parts of Staffordshire have been affected, with several centimetres of
snow reported in Flash, England's highest village.
Commuters have been urged to allow extra journey time,
and the Met Office has issued snow and ice warnings.
More on the snow and other stories in West Yorkshire Weather
updates for Lancashire and Greater Manchester BBC Weather
presenter Kay Crewdson said conditions were due to slowly
improve into Saturday. Molly Greenwood reported 10cm of snow
in the Huddersfield area. "Don't think I'm going anywhere," she said.
Zulfi Hussain said the snow was causing "traffic chaos" in Woodhall Road,
Calverley, near Leeds. Elliott Hudson, another West Yorkshire resident,
said: "Looks like I have woken up in Narnia."
West Yorkshire's Liversedge FC, who have had to cancel every home
game for the last four months due to bad weather,
tweeted a picture of snow with the caption:
"It's not looking good for Liversedge FC's home game with Worksop Town tomorrow."
The A628 Woodhead, A57 Snake Pass and A537 Cat and
Fiddle roads are all affected, with delays reported on
the M65 motorway. Highways England said the A57 eastbound
in Great Manchester is closed between M67/A560 and B6174
due to severe weather conditions. It said teams were working
to clear the road. Tony Hallwood, from Leeds Bradford Airport,
said it reopened at about 09:00 GMT after crews used ploughs
to clear snow from the runway. He said: "We are asking passengers
to make their way to the airport as early as they can given the difficult
conditions." Bus operators are also reporting delays to all services
across West Yorkshire. Oldham Council has said 48 schools had closed
this morning as a result of the snow and severe weather.
Drivers are also being asked to take extra care after snow
fell overnight in some parts of Northern Ireland.
A Met Office yellow warning for ice and snow in
northern England and Wales ended at 15:00.

**GT:** Heavy snowfall has caused travel disruption in parts of northern England.

**CLAPS:** Flights have been disrupted after a large avalanche hit parts of England.

**Article**: But once the votes are counted, what can residents expect to pay in council tax?
Below are the figures for a Band D property for every council area
in Wales for the current financial year of 2017/18,
how much that has gone up by for the current year,
and what the average property in the area actually pays.
They are grouped here by police force region -
council tax includes the police precept which is added to
the overall bill paid by homes. Local government is not fully
funded by council tax. Much of the funding for councils comes
in the form of grants from the Welsh Government,
which in turn gets its funding from the UK government in London.
In 2017/18 a total of £4.1bn is being divided among Wales' 22 councils.
The lions share of council cash goes on schools
- with social services following behind, as shown in the graph above.
Residents pay council tax based on which band their property is in,
based on its worth. Band D has historically been used as
the standard for comparing council tax levels between and across local
authorities. It is used to charge tax to a property that, in Wales,
was worth between £91,001 to £123,000 on April 2003 values.
Council tax gets lower the cheaper a property is,
and higher the more expensive a property is.
Council tax figures source: Welsh Government

**GT:** Voters will go to the polls on Thursday to determine who will represent them on local councils.

**CLAPS:** The people of Wales are voting in a referendum on whether or not to pay council tax.

**Article:** The side's appearance in France will be its first at a major
tournament since the 1958 World Cup.
Players and coaches left their base at the Vale Resort,
Vale of Glamorgan, on Saturday and headed to Cardiff Airport.
After a send-off from pupils from Ysgol Treganna, Cardiff,
the team took off for a friendly in Sweden on Sunday.
They will then head to France ahead of the team's first
game of the tournament against Slovakia on 11 June.

**GT:** Wales' football team has departed the country as their Euro 2016 preparations reach a climax.

**CLAPS:** Wales' Euro 2016 squad have arrived in France for the first time since 1958.

**Article:** The 40-year-old, from the South Bank area of Teesside,
was discovered on the A66 in the early hours "in a distressed state"
with wounds to his groin after the attack.
The road, from Greystones Roundabout to Church Lane in Middlesbrough,
was shut earlier while searches of the area were carried out.
It has now reopened. A 22-year-old man was arrested on suspicion of assault
and later bailed. Cleveland Police said the injured man had been
placed in an induced coma in hospital. The force said in a statement:
"Police can confirm that the man found this morning on the A66
had wounds to his groin area. "Officers are continuing to
investigate and are appealing for anyone with information to contact them."

**GT:** A man has been found by the side of a road with his penis cut off.

**CLAPS:** A man is in an induced coma after being found with serious injuries on a Teesside road.

**Article**: In July, a major bug was discovered in the software
that could let hijackers access data on up to a billion phones.
Manufacturers have been slow to roll out a fix because many variations
of Android are widely used. One Android expert said it was "about time"
phone makers issued security fixes more quickly.
Android has been working to patch a vulnerability, known as Stagefright,
which could let hackers access a phone's data simply by sending somebody
a video message. "My guess is that this is the single largest software
update the world has ever seen," said Adrian Ludwig,
Android's lead engineer for security, at hacking conference Black Hat.
LG, Samsung and Google have all said a number of their handsets
will get the fix, with further updates every month.
Android is an open source operating system, with the software
freely available for phone manufacturers to modify and use on their handsets.
The Google-led project does provide security fixes for the software,
but phone manufacturers are responsible for sending the updates
to their devices. Some phones running old versions of Android
are no longer updated by the manufacturer.
Many companies also deploy customised versions of Android
which take time to rebuild with the security changes.
Apple and BlackBerry can patch security problems more quickly
because they develop both the software and the hardware for their devices.
BlackBerry's software is reviewed by mobile networks before being sent to
handsets, while Apple can push updates to its phones whenever it wants.
"The very nature of Android is that manufacturers
add their own software on top, so there have been delays
in software roll-outs," said Jack Parsons, editor of Android Magazine.
"In the US it's even worse because mobile carriers often add their
own software too, adding another layer of bureaucracy holding up security fixes.
"There's no real villain here, that's just how the system works.
But there will always be security concerns with software,
so it's right that some of the manufacturers are stepping up to deal with this now."

**GT:** Samsung, LG and Google have pledged to provide monthly security
updates for smartphones running the Android operating system.

**CLAPS:** The world's largest software update is to be issued by Google-led Android.

**Article:** The move follows a claim by Crossmaglen Rangers player
Aaron Cunningham that he was the victim of verbal abuse during
the 2 December Ulster football final. The Ulster Council carried out an
investigation and BBC Sport understands one Kilcoo player is to
be banned for six months and another for four months.
Kilcoo said they had not been notified, and the players could appeal.
The two suspensions have yet to be officially confirmed
by the Ulster Council. It is believed the case was the
first time an allegation of racial abuse had been lodged
with the provincial governing body. When an investigation was announced,
Ulster GAA president Aogán O Fearghail, said anyone found
guilty of racism would be dealt with severely.
Kilcoo released a statement saying the club condemned
abuse and would co-operate with the Ulster Council's investigation.
The Gaelic Athletic Association, which governs the sport in Ireland,
is to discuss how to deal with racism at its annual congress in March.

**GT:** Two Kilcoo players are to be suspended by Ulster GAA chiefs following allegations of racial abuse.

**CLAPS:** Two Kilcoo players have been suspended by the Ulster GAA for alleged racial abuse.

Table 6: Generated Questions by CLAPS from SQuAD. Answer spans are highlighted.

**Context**: ... The Broncos finished the regular season with a 12-4 record, and denied the New England Patriots a chance to defend their title from Super Bowl XLIX by defeating them 20-18 in the AFC Championship Game. They joined the Patriots, Dallas Cowboys, and Pittsburgh Steelers as one of four teams that have made **eight** appearances in the Super Bowl.

**GT:** How many appearances have the Denver Broncos made in the Super Bowl?

**CLAPS:** How many Super Bowl appearances have the Broncos made?

**Context:** In late November 2015, reports surfaced stating that "multiple acts" would perform during the halftime show. On December 3, the league confirmed that the show would be headlined by the **British** rock group Coldplay. On January 7, 2016, Pepsi confirmed to the Associated Press that Beyoncé, who headlined the Super Bowl XLVII halftime show and collaborated with Coldplay on the single "Hymn for the Weekend", would be making an appearance. Bruno Mars, who headlined the Super Bowl XLVIII halftime show, and Mark Ronson also performed.

**GT:** What nationality is the band Coldplay?

**CLAPS:** What nationality was Coldplay?

**Context:** There are 13 natural reserves in Warsaw - among others, Bielany Forest, Kabaty Woods, Czerniaków Lake. About 15 kilometres (9 miles) from Warsaw, the Vistula river's environment changes strikingly and features a perfectly preserved ecosystem, with a habitat of animals that includes the **otter, beaver and hundreds of bird species**. There are also several lakes in Warsaw - mainly the oxbow lakes, like Czerniaków Lake, the lakes in the Łazienki or Wilanów Parks, Kamionek Lake. There are lot of small lakes in the parks, but only a few are permanent - the majority are emptied before winter to clean them of plants and sediments.

**GT:** What animals does the Vistula river's ecosystem include?

**CLAPS:** What animals are included in the Vistula river's habitat?

**Context:** "The FSO Car Factory was established in 1951. A number of vehicles have been assembled there over the decades, including the Warszawa, Syrena, Fiat 125p (under license from Fiat, later renamed FSO 125p when the license expired) and the Polonez. The last two models listed were also sent abroad and assembled in a number of other countries, including Egypt and Colombia. In 1995 the factory was purchased by the South Korean car manufacturer Daewoo, which assembled the Tico, Espero,Nubia, Tacuma, Leganza, Lanos and Matiz there for the European market. In 2005 the factory was sold to **AvtoZAZ**, a Ukrainian car manufacturer which assembled there the Chevrolet Aveo. The license for the production of the Aveo expired in February 2011 and has since not been renewed. Currently the company is defunct."

**GT:** Who bought the factory in 2005?

**CLAPS:** To whom was the factory sold in 2005?

**Context:** The Scotland Act 1998, which was passed by the Parliament of the United Kingdom and given royal assent by Queen Elizabeth II on 19 November 1998, governs the functions and role of the Scottish Parliament and delimits its legislative competence. The Scotland Act 2012 extends the **devolved competencies**. For the purposes of parliamentary sovereignty, the Parliament of the United Kingdom at Westminster continues to constitute the supreme legislature of Scotland. However, under the terms of the Scotland Act, Westminster agreed to devolve some of its responsibilities over Scottish domestic policy to the Scottish Parliament. Such devolved mattersïnclude education, health, agriculture and justice. The Scotland Act enabled the Scottish Parliament to pass primary legislation on these issues. A degree of domestic authority, and all foreign policy, remain with the UK Parliament in Westminster. The Scottish Parliament has the power to pass laws and has limited tax-varying capability. Another of the roles of the Parliament is to hold the Scottish Government to account.

**GT:** What does the Scotland Act of 2012 extend?

CLAPS: What does the Scotland Act 2012 extend?

**Context:** Stage 1 is the first, or introductory stage of the bill,
where the minister or member in charge of the bill will formally introduce it to Parliament together
with its accompanying documents-Explanatory Notes, a Policy Memorandum
setting out the policy underlying the bill,
and a Financial Memorandum setting out the costs and savings associated with it.
Statements from the Presiding Officer and
the member in charge of the bill are also lodged indicating whether the bill is
within the legislative competence of the Parliament.
Stage 1 usually takes place, initially, in the relevant committee or committees
and is then submitted to **the whole Parliament**
for a full debate in the chamber on the general principles of the bill.
If the whole Parliament agrees in a vote to the general principles of the bill, it then proceeds to Stage 2.

**GT:** Where are bills typically gestated in Stage 1?

**CLAPS:** Where does Stage 1 usually take place?

**Context:** Moderate and reformist Islamists who accept and
work within the democratic process include parties like the Tunisian Ennahda Movement.
Jamaat-e-Islami of Pakistan is basically a socio-political
and democratic Vanguard party but has also gained political influence
through military coup d'état in past.
The Islamist groups like Hezbollah in Lebanon and Hamas in **Palestine** participate
in democratic and political process as well as armed attacks,
seeking to abolish the state of Israel.
Radical Islamist organizations like al-Qaeda and the Egyptian Islamic Jihad,
and groups such as the Taliban, entirely reject democracy,
often declaring as kuffar those Muslims who support it (see takfirism),
as well as calling for violent/offensive jihad
or urging and conducting attacks on a religious basis.

**GT:** Where does Hamas originate?

**CLAPS:** Where are Hamas located?

**Context:** Sayyid Abul Ala Maududi was an important early twentieth-century figure
in the Islamic revival in India, and then after independence from Britain, in Pakistan.
Trained as a lawyer he chose the profession of journalism,
and wrote about contemporary issues and most importantly about Islam and Islamic law.
Maududi founded the Jamaat-e-Islami party in 1941 and remained its leader until 1972.
However, Maududi had much more impact **through his writing**
than through his political organising. His extremely influential books (translated into many languages)
placed Islam in a modern context, and influenced not only conservative ulema
but liberal modernizer Islamists such as al-Faruqi,
whose Ïslamization of Knowledgeċarried forward some of Maududi's key principles.

**GT:** Where did Maududi exert the most impact?

**CLAPS:** How did Maududi have more impact on Islam than his political organising?

**Context:** ByLike many other mainline Protestant denominations in the United States,
the United Methodist Church has experienced significant membership losses in recent decades.
At the time of its formation, the UMC had about 11 million members
in nearly **42,000** congregations.
In 1975, membership dropped below 10 million for the first time.
In 2005, there were about 8 million members in over 34,000 congregations.
Membership is concentrated primarily in the Midwest and in the South.
Texas has the largest number of members, with about 1 million.
The states with the highest membership rates are
Oklahoma, Iowa, Mississippi, West Virginia, and North Carolina.

**GT:** At the time of its formation, how many congregations did the UMC have?
**CLAPS:** How many congregations did the UMC have at the time of its formation?

**Context:** Celoron's expedition force consisted of about 200 Troupes de la marine and 30 Indians.
The expedition covered about 3,000 miles (4,800 km) between June and November 1749.
It went up the St. Lawrence, continued along the northern shore of Lake Ontario,
crossed the portage at Niagara, and followed the southern shore of Lake Erie.
At the Chautauqua Portage (near present-day Barcelona, New York),
the expedition moved inland to the Allegheny River, which it followed to the site of present-day Pittsburgh.
There Céloron buried lead plates engraved with the French claim to the Ohio Country.
Whenever he encountered British merchants or fur-traders,
==Celoron informed them of the French claims on the territory and told them to leave.==

**GT:** How did Celeron handle business on trip?

**CLAPS:** What did Celoron do when he encountered the British?

**Context:** Like many cities in Central and Eastern Europe, infrastructure in Warsaw
suffered considerably during its time as an Eastern Bloc economy
- though it is worth mentioning that the initial Three-Year Plan
to rebuild Poland (especially Warsaw) was a major success,
but what followed was very much the opposite.
However, over the past decade Warsaw has seen many improvements
due to solid economic growth, an increase in foreign investment
as well as funding from the European Union. In particular,
the city's metro, roads, sidewalks, health care facilities
and sanitation facilities have ==improved markedly==. answer:improved markedly

**GT:** Warsaw's sidewalks and sanitation facilities are some examples of things which have what?

**CLAPS:** What has happened to Warsaw's infrastructure in the past decade?

**Context:** Several commemorative events take place every year.
Gatherings of ==thousands== of people on the banks of the Vistula
on Midsummer's Night for a festival called Wianki (Polish for Wreaths)
have become a tradition and a yearly event in the programme of cultural events
in Warsaw. The festival traces its roots to a peaceful pagan ritual
where maidens would float their wreaths of herbs on the water to predict
when they would be married, and to whom.
By the 19th century this tradition had become a festive event,
and it continues today. The city council organize concerts and other events.
Each Midsummer's Eve, apart from the official floating of wreaths,
jumping over fires, looking for the fern flower,
there are musical performances, dignitaries' speeches, fairs
and fireworks by the river bank.

**GT:** How man people gather along the banks of the Vistula for the Wianki festival?

**CLAPS:** How many people gather on the banks of the Vistula on Midsummer's Night
for a festival called Wianki?

**Context:** The origin of the legendary figure is not fully known.
The best-known legend, by Artur Oppman, is that long ago two of Triton's
daughters set out on a journey through the depths of the oceans and seas.
One of them decided to stay on the coast of Denmark and can be seen sitting
at the entrance to the port of Copenhagen.
The second mermaid reached the mouth of the Vistula River and
plunged into its waters. She stopped to rest on a sandy beach
by the village of Warszowa, where fishermen came to admire her beauty
and listen to her beautiful voice. A greedy merchant also heard her songs;
he followed the fishermen and ==captured== the mermaid.

**GT:** What did a greedy merchant do to the mermaid?

**CLAPS:** What did Oppman do to the mermaid?

**Context:** Warsaw remained the capital of the Polish-Lithuanian
Commonwealth ==until 1796==, when it was annexed by the Kingdom of Prussia
to become the capital of the province of South Prussia.
Liberated by Napoleon's army in 1806, Warsaw was made
the capital of the newly created Duchy of Warsaw.
Following the Congress of Vienna of 1815, Warsaw became the centre of
the Congress Poland, a constitutional monarchy under a personal union
with Imperial Russia. The Royal University of Warsaw was established in 1816.

**GT:** How long was Warsaw the capital of the Polish-Lithuanian Commonwealth?

**CLAPS:** How long did Warsaw remain the capital of the Polish-Lithuanian Commonwealth?

---

**Context:** John Paul II's visits to his native country in 1979
and 1983 brought support to the budding solidarity movement
and encouraged the ==growing anti-communist== fervor there.
In 1979, less than a year after becoming pope, John Paul celebrated
Mass in Victory Square in Warsaw and ended his sermon with a call to
"renew the face" of Poland: Let Thy Spirit descend!
Let Thy Spirit descend and renew the face of the land!
This land! These words were very meaningful for the Polish citizens
who understood them as the incentive for the democratic changes.
**GT:** What is St. John's Cathedral an example of, stylistically?

**CLAPS:** St. John's Cathedral is a typical example of what style?

---

**Context:** Gothic architecture is represented in the majestic churches
but also at the burgher houses and fortifications.
The most significant buildings are St. John's Cathedral (14th century),
the temple is a typical example of the so-called ==Masovian gothic style==,
St. Mary's Church (1411), a town house of Burbach family (14th century),
Gunpowder Tower (after 1379) and the Royal Castle Curia Maior (1407Ž0131410).
The most notable examples of Renaissance architecture in the city
are the house of Baryczko merchant family (1562), building called "The Negro"
(early 17th century) and Salwator tenement (1632). The most interesting examples
of mannerist architecture are the Royal Castle (1596Ž0131619) and
the Jesuit Church (1609Ž0131626) at Old Town.
Among the first structures of the early baroque the most important
are St. Hyacinth's Church (1603-1639) and Sigismund's Column (1644).

**GT:** What is St. John's Cathedral an example of, stylistically?

**CLAPS:** St. John's Cathedral is a typical example of what style?

Table 7: Translation of Romanian by CLAPS from WMT'16 RO-EN.

**RO:** De partea cealaltă, 47% dintre alegătorii republicani afirmă că ar fi „nemulțumiți" sau „supărați" dacă favoritul Jeb Bush câștigă cursa pentru nominalizare.

**GT :** On the flip side, 47 percent of Republican voters say they would be "dissatisfied" or "upset" if establishment favorite Jeb Bush becomes the nominee.

**CLAPS**: On the other hand, 47% of Republican voters say they would be "unsatisfied" or "uneasy" if the winner Jeb Bush wins the nomination race.

**RO:** Datoria va deveni o problemă importantă.

**GT:** Debt will become a big issue.

**CLAPS:** Debt will become an important issue.

**RO:** Se pare că Moreno a avut noroc și a scăpat de cartonașul roșu și de concesia unei lovituri de pedeapsă.

**GT:** Moreno appeared fortunate to escape a red card and the concession of a spot-kick.

**CLAPS:** It seems that Moreno has been lucky and escaped the red card and the concession of a penalty kick.

**RO:** Astfel de decizii nu se iau niciodată ușor.

**GT:** Such decisions are never taken lightly.

**CLAPS:** Such decisions are never made easily.

**RO:** Toate astea au cu certitudine un cost.

**GT:** All that stuff sure does take a toll.

**CLAPS:** All of this certainly has a cost.

**RO:** Astfel de decizii nu se iau niciodată ușor.

**GT:** Such decisions are never taken lightly.

**CLAPS:** Such decisions are never made easily.

**RO:** Deci, nu am mai gasit niciun sens de a continua pe acest drum.

**GT:** So, I had no reason to continue on this path.

**CLAPS:** So there is no point in continuing on this road.

**RO:** Profitul a crescut cu 5%, până la 12,3 miliarde dolari.

**GT:** Revenue rose 5 percent, to $12.3 billion.

**CLAPS:** Profit has increased by between 5% and $12.3 billion.

**RO:** „Este imposibil ca toată lumea să vină în Europa" - Dalai Lama - RT news

**GT:** "It is impossible for everyone to come to Europe" - Dalai Lama - RT News

**CLAPS:** "It is impossible for everyone to come to Europe" - the Dalai Lama - RT news

**RO:** Spune povestea deținutului așa cum își dorește acesta, obține acces.

**GT:** Tell the inmate's story the way he likes it, get access.

**CLAPS:** It tells the detainee's story as he wants, it gets access.

**RO:** Peter Moody a afirmat că stewarzii Racing Victoria sau încercat să infiltreze un spion la grajdurile sale anul trecut și a amenințat că se retrage imediat din curse.

**GT:** Peter Moody has alleged Racing Victoria stewards attempted to plant a spy in his stables last year and threatened to quit racing immediately.

**CLAPS:** Peter Moody said stewarzia Racing Victoria or tried to infiltrate a spy at her track last year and threatened to withdraw immediately from the race.

**RO:** Corbyn, Tsipras și Syriza în Grecia, Podemos în Spania, chiar Bernie Sanders în Statele Unite își alimentează retorica populistă din frustrările acumulate în societățile oocidentale.

**GT:** Corbyn, Tsipras and Syriza in Greece, Podemos in Spain, even Bernie Sanders in the United States feed their populist rhetoric with the frustrations accumulated in the Western world.

**CLAPS:** Corbyn, Tsipras and Syriza in Greece, Podemos in Spain, even Bernie Sanders in the United States are fuelling their populist rhetoric from the frustrations gained in the occident societies.

---

**RO:** Pentru România, chiar și cota voluntară propusă de București, în cuantum de circa 1500 de suflete, ne depășește cu mult bunele intenții de solidaritate cu Uniunea Europeană exprimate în ultimele luni.

**GT:** For Romania, even the voluntary quota proposed by Bucharest, amounting to about 1,500 souls, surpasses by far our good intentions of solidarity with the European Union expressed in recent months.

**CLAPS:** For Romania, even Bucharest's proposed voluntary quota, amounting to around 1 500 souls, goes far beyond our good intentions of solidarity with the European Union expressed in recent months.

---

**RO:** 7.000 de euro pe an pentru a închiria Clubul Pogor.

**GT:** 7,000 Euro per year to rent Pogor Club.

**CLAPS:** 7,000 euros per year to rent the Pogor Club.

---

**RO:** Crina a fost internată împreună cu bunica ei la spitalul municipal din Pașcani, iar tot atunci tatăl fetei, Costică Balcan, a plecat la muncă în orașul Alexandria, județul Teleorman.

**GT:** Crina and her grandmother were admitted to the hospital of the municipality of Pașcani, and on the same day her father, Costica Balcan went to work in the city of Alexandria, Teleorman County.

**CLAPS:** The crisis was admitted with her grandmother to the municipal hospital in Pascani, and the girl's father, Costică Balcan, also went to work in the town of Alexandria, Teleorman County.

---

**RO:** În țară, părinții Crinei, Alina și Costică, au continuat separat căutările, fiecare pe unde a putut.

**GT:** I was just a little girl when I found out.

**CLAPS:** I was small when I found out.

---

**RO:** Cu o săptămână înainte de întâlnire nici nu prea mai putea să doarmă.

**GT:** A week before the meeting he hadn't really been able to sleep.

**CLAPS:** A week before the meeting, we could not even sleep much longer.

---

**RO:** "Ca să îți vezi copilul după atâția ani, este ceva", ne spune o nepoată, venită și ea cu Costică de la Ruginoasa.

**GT:** "It's something extraordinary to see your baby after so many years", says a niece who came with Costică from Ruginoasa.

**CLAPS:** "To see your child after so many years, it is something," says a little girl, who also comes with Costica from Ruginoasa.

---

**RO:** În Piața Unirii, sub cerul începutului de toamnă, familia din Ruginoasa își unește din nou destinul cu fata lor din Palermo.

**GT:** In Piața Unirii, beneath the autumn sky, the family from Ruginoasa joins their destiny with their daughter in Palermo.

**CLAPS:** In the Square of the Union, under the skies of the early autumn, the family of Ruginoasa is once again uniting their destiny with their mother in Palermo.

---

**RO:** "Sper să am o amintire frumoasă cu ei, pentru că acum sunt atât de multe lucruri de spus", ne mărturisește Crina.

**GT:** "I hope to gain a beautiful memory after meeting them, because now there are so many things to say," confesses Crina.

**CLAPS:** "I hope to have a nice memory with them, because there is so much to say now," Crina tells us.

---

**RO:** În weekendul care a trecut, Alina și Costică și-au văzut pentru prima dată fata pierdută pe holurile spitalului din Iași, în urmă cu 20 ani.

**GT:** Last weekend, Alina and Costică saw the girl who was lost on the halls of the hospital in Iasi for the first time in 20 years.

**CLAPS:** For the first time in the weekend, Alina and Costica saw their girl lost in hospital rooms in Jasmine 20 years ago.

---

**RO:** "A durat până la urmă cam o lună de zile până să îl găsim pe tatăl biologic al fetei", ne explică comisarul șef Romică Ichim.

**GT:** Then they gave me some pointers, only to find that the person I was directed to wasn't he person I was looking for.

**CLAPS:** They gave me some indications then, just that the person found was not the one wanted.

