# OpenReview forum: "Contrastive  Learning  with Adversarial Perturbations for Conditional Text Generation"
_ICLR.cc/2021/Conference — ICLR 2021 Poster_

### Official Review · AnonReviewer3 · 2020-10-29
**Interesting technique, but significance of the improvements over baselines unclear**

**Rating:** 6
**Confidence:** 3

**Review:**

# Summary
Proposes contrastive learning method for conditional text-generation. Here we maximize similarity (of representations) between source and target sequences (positive) while minimizing similarity with false targets (negative). Additional positives and negatives are created in the sequence representation space by adding perturbations to decoder (output) hidden states to minimize/maximize conditional likelihood p(y|x). It is shown this works a lot better than the naive contrastive approach of sampling random non-target sequences.

The full model is based on T5-small (Raffel et al) and combines contrastive objective with regular MLE objective by simple addition. Modest improvements over T5-small are observed on Translation, Summarization, and Question-generation seq2seq tasks.

# Pros
1. Diversity of seq2seq tasks, with consistent improvements over baseline T5-MLE (small).
2. Possibly improves the exposure bias issue of regular MLE seq2seq training.
3. Complementary to seq2seq MLE training and can be used to improve it in general, not just text generation.

# Cons
1. The improvements are consistent but appear to be modest. It is unclear whether the improvements would persist on the larger T5 model sizes. Would it be possible to study this (e.g. medium size)?
2. Please add SOTA results in the tables for the various tasks for reference.
3. Please discuss effect on training/inference speed.
4. Since this is generation, more non-cherry-picked example decodes would be informative to have in the appendix.
5. Even better would be some basic human evaluation of generated outputs to verify whether meaningful quality improvements are made.
6. Scheduled Sampling (Bengio et al) should be discussed and perhaps compared as it is a well-known method for addressing exposure bias.
7. Should discuss relationship to Virtual Adversarial Training (Miyato et al)

# Clarifications
1. Are all the models initialized with T5-MLE or are they trained from scratch on C4 for the same number of steps as T5-MLE?

---

> ### Author Response · Authors · 2020-11-16
> **Response to R3**
>
> We sincerely appreciate your constructive comments. We respond to your main concerns below:
>
> ---
>
> Q1.The improvements are consistent but appear to be modest. It is unclear whether the improvements would persist on the larger T5 model sizes. Would it be possible to study this (e.g. medium size)?
> - We have **conducted experiments on a larger T5 model**as requested (Table below). Due to the limited resources, we have only performed experiment on machine translation with smaller batch size. Yet, our method consistently improves the performance in machine translation. This is somehow expected since our method is model-agnostic, but we believe that this result will further strengthen our paper as it shows the generality of our method.
>
> |         | BLEU-1 | BLEU-2 | BLEU-3 | BLEU-4 | Sacre-BLEU |
> |---------|--------|--------|--------|--------|------------|
> | T5-small | 57.76  | 44.45  | 35.12  | 28.21  | 32.43      |
> | +CLAPS | **58.98** | **45.72**  | **36.39**  | **29.41**  | **33.96**     |
> |---------|--------|--------|--------|--------|------------|
> | T5-base | 58.57  | 45.59  | 36.47  | 29.66  | 33.96      |
> | +CLAPS  | **59.58**  | **46.82**  | **37.70**  | **30.85**  | **35.18**    |
>
> ---
>
>
> Q2.Please add SOTA results in the tables for the various tasks for reference.
> - We have included the SOTA results in the revised version of the paper **(Table 1 and 2)**.
>
> ---
>
> Q3.Please discuss effect on training/inference speed.
> - For inference, it does not require any extra computation. Additional computation for adversarial perturbations is marginal.  Compared to MLE training, it takes extra 15 minutes for every epoch in machine translation.
>
> ---
>
> Q4.Since this is generation, more non-cherry-picked example decodes would be informative to have in the appendix.
> - We have included more examples in the **Page 16-24**of the Appendix.
>
> ---
>
> Q5.Even better would be some basic human evaluation of generated outputs to verify whether meaningful quality improvements are made.
> - As suggested, we have conducted a pairwise **human evaluation of the sentences (summaries or questions) generated** by CLAPS and the baseline (Page 9, paragraph "Human Evaluation"). For text summarization, **70%** of human annotators choose the sequences generated by our model is better than the baseline. For question generation, **85%** of them say the quality of the sentences from our model is better than that of the baseline.
>
> ---
>
> Q6.Scheduled Sampling (Bengio et al) should be discussed and perhaps compared as it is a well-known method for addressing exposure bias.
>
> - Thanks for the reference and we have included the discussion of the Scheduled Sampling in the **related work section of the revision**. However if we use it, we cannot benefit from the advantage of Transformer architecture, which enables parallel computing, because it sequentially generates a token and feeds it into the mode as input of next time step.
>
> ---
>
> Q6.Should discuss relationship to Virtual Adversarial Training (Miyato et al)
>
> - The goal of our work is totally different from [2]. Although how we add perturbation is similar, [2] utilizes the adversarial examples as augmented data, and enforce consistency across the predictions across original unlabeled example and its perturbation, for semi-supervised learning of a text classification model. However, we leverage the perturbed representation of target sentences as negative examples for contrastive learning and push those examples far apart from the source sentences. As a result, the embedding of the target sentence is close to the source embedding, so the seq2seq model can generate more quality sentences. To compare the methods which leverage regularization technique to induce the smoothness of function, we implement the method "R3F" [3] and report its performance. Please see the result in the common response, and results in **Table 1 and 2 of the revision**.
>
> ---
>
> Q7. Are all the models initialized with T5-MLE or are they trained from scratch on C4 for the same number of steps as T5-MLE?
> - All the methods using T5 model, except scratch-T5-MLE and scratch-CLAPS, are initialized with T5 model pretrained on C4 and trained for 20 epochs.
>
> ---
>
> References
>
> [1] Bengio, Samy, et al. "Scheduled Sampling for Sequence Prediction with Recurrent Neural Networks." NIPS. 2015.
>
> [2]  Miyato, Takeru, Andrew M. Dai, and Ian Goodfellow. "Adversarial training methods for semi-supervised text classification." arXiv preprint arXiv:1605.07725 (2016).
>
> [3] Aghajanyan, Armen, et al. "Better fine-tuning by reducing representational collapse." arXiv preprint arXiv:2008.03156 (2020).

---

> ### Author Response · Authors · 2020-11-24
> **Summary of Response**
>
> This is a summary of what we have provided in response to your feedback
> - We have added discussion about VAT and compare ours against its variants R3F.
> - We have performed additional experiments with the randomly initialized Transformer and report the results.
> - We report the result of the larger T5-model for machine translation.
> - We have conducted human evaluation to qualitatively validate the improvement of our proposed method.
> - We have included more examples in Appendix.
> - We have included Scheduled sampling in the related section of the revision.
> - We have described the effect of our proposed method on training and inference speed.
> - We have clarified the experimental setup.
> - We have included the SOTA results in the revised version of the paper.
>
> We have done our best to address every comment in the initial feedback. The end of the author discussion period is approaching, so if there are any further questions or concerns we would like to do our best to address them as soon as possible.
>
> Thank you for your feedback

---

### Official Review · AnonReviewer4 · 2020-10-29

**Rating:** 5
**Confidence:** 4

**Review:**

========================

Paper Summary:

This paper proposes to add contrastive learning to the sequence-to-sequence generation problem. More specifically, the authors apply a contrastive loss on the globally pooled hidden representation of the generated hidden states. The key novelty is to apply adversarial gradients to obtain both hard negative and hard positive examples. The proposed method can improve a state-of-art pretrained transformer model (T5) on 3 tasks: machine translation (WMT16 En-Ro), abstractive summarization (XSum), and question generation (SQuAD).

==========================

Overall review

Although the proposed method seems to be effective and new, the concerns outweighs the contributions in my opinion. I am leaning towards rejection for now. Please try to address my concerns during the rebuttal period.

Pros

-	The idea of using adversarial gradients to generate hard negative/positive is novel, at least for contrastive learning and sequence generation problems.
-	Improvement is demonstrated on a strong pre-trained transformer model (T5).
-	This method is experimented on 3 tasks and could possibly be extended to any seq2seq generation.

Cons

-	The sdfgclaim of solving the ‘exposure bias’ is somewhat exaggerated.
-	The proposed method is somewhat straight-forward and lacking theoretical insights/guarantees.
-	The method is only applied on a small version of T5, which is limited. How about other pretrained (potentially larger) models? How about non-pretrained models such as randomly initialized Transformers/LSTMs?

==========================

Detailed Review

The authors claimed to mitigate the exposure bias problem for sequence generation. However, the original `exposure bias’ problem refers to not seeing incorrectly generated text tokens as training input, which leads to train-test mismatch. In this work, the model does not see any self-generated negative tokens as input, but only pooled adversarial hidden states. It does not mitigate train-test mismatch at all. Therefore, the current presentation may be misleading. It might benefit the paper to also compare and contrast to adversarial training for NLU such as SMART/FreeLB.

Moreover, this work does not provide new theoretical insights. The hard negatives/positives do not have theoretical guarantee. It is not clear to me why Eqn (6) & (7) will be a distant positive. If g = f, then H = H_bar. Moreover, in MT and Sum., adversarial step size eta and epsilon are set the same. This is inconsistent with the intuition of near negative and distant positive claimed in the paper.


==========================

Other Questions / Suggestions

-	In Eqn. (2), why not use 2-layer MLP as in SimCLR?
-	In experiments, maybe add a CLAPS w/o negative so that readers would know which is more important.
-	Why not train as how you generate Table 3 example? This will then better solve the train-test mismatch (exposure bias), although maybe at a cost of slow training.
-	Some human evaluation on a larger set of generated examples would help. For example, how many hard negatives are actually being recognized as negative by human.

---

> ### Author Response · Authors · 2020-11-16
> **Response to R4-(1/2)**
>
> Thank you for the helpful suggestion. We have included detailed discussions about the suggested references in the revision and provide responses to your comments below:
>
> ---
>
> > Q1. The  claim of solving the ‘exposure bias’ is somewhat exaggerated.
>
> - We will **tone down**our claim as follows: "our proposed method aims toward mitigating the problem of teacher forcing by generating positive and negative examples".
>
> ---
>
> > Q2. The proposed method is somewhat straight-forward and lacking theoretical insights/guarantees.
>
> - Our method is **not straightforward**. The baseline which straight-forwardly applies contrastive learning (**T5-MLE-contrastive**) yields worse performance than the T5-MLE. Moreover, existing contrastive learning framework for seq2seq learning (T5-SSMBA, T5-WordDropout Contrastive) result in marginal performance improvements.
>
> - We clearly motivated the challenge in contrastive learning with pretrained language models in Figure 2, where negative examples generated are too easy, and proposed a principled way to automatically generate effective positive and negative examples for contrastive learning of seq2seq models.
>
> - We generate negative examples (imposters) by adding perturbation to the hidden representation of target sentences such that its embedding is close to the original embedding but its semantics are drastically different. On the other hand, we generate hidden representations of additional positive examples (distant targets) by maximizing cosine distance but keep its conditional log likelihood high. The extra computational cost is a little and we empirically show that our method is consistently effective across three different text generation tasks.
>
> ---
>
> > Q3. How about other pretrained (potentially larger) models? How about non-pretrained models such as randomly initialized Transformers/LSTMs?
>
> - We performed the experiments with a **randomly initialized Transformer**that have the identical architecture as T5, and have reported the results in the main response and the revision **(Table 1 and 2)**.
>
> ---
>
> > Q4. The authors claimed to mitigate the exposure bias problem for sequence generation. However, the original `exposure bias’ problem refers to not seeing incorrectly generated text tokens as training input, which leads to train-test mismatch.
> In this work, the model does not see any self-generated negative tokens as input, but only pooled adversarial hidden states. It does not mitigate train-test mismatch at all. Therefore, the current presentation may be misleading.
>
> - In contrast to teacher forcing where the model is trained with only ground truth tokens, with our proposed method,  seq2seq model is exposed to various hidden representations of incorrect sentences, including hard negative examples constructed by adversarial perturbation as in Eq.(3). By contrasting ground truth or generated positive pair and pair of source and negative examples (imposters), the model learns better representation of target sentences, which leads to better quality of target sentence generation.
>
> ---
>
> > Q5. Compare and contrast to adversarial training for NLU such as SMART/FreeLB.
>
> - SMART/FreeLB leverages adversarial training to finetune the pretrained language model to induce the smoothness of classifier and prevent the overfitting. However, they require additional extra forward and backward steps, which is computationally prohibitive.
>
> - Instead, we compare against a similar method **(R3F) [2]**, which is faster, outperforms SMART and FreeLB, and is applicable to the conditional text generation. The results show that ours significantly **outperforms R3F**. Please see the global response and the revision.
>
> ---

---

> > ### Author Response · Authors · 2020-11-16
> > **Response to R4-(2/2)**
> >
> > > Q6. Not clear why Eqn (6) & (7) will be a distant positive. If g = f, then H = H_bar. Moreover, in MT and Sum., adversarial step size eta and epsilon are set the same. This is inconsistent with the intuition of near negative and distant positive claimed in the paper.
> >
> > - $\eta$ and $\epsilon$ are hyperparameters which control the strength of perturbation.
> >
> > - Eq.6 enforces the perturbed embedding to be distant from the source sentence embedding, because the perturbation should minimize the contrastive objective which maximizes the cosine distance between source and target sentences.
> >
> > - On the other hand, Eq.7 prevents the semantics of perturbed embedding from deviating from the original target sentence by minimizing the KL-divergence between the original conditional distribution and the conditional distribution after the perturbation. This will yield perturbed embeddings that are distant from the original embedding but with similar semantics.
> >
> > ---
> >
> > > Q7. In Eqn. (2), why not use 2-layer MLP as in SimCLR?
> >
> > - There is no performance difference between a 1-layer and 2-layer MLP, so we use the former to reduce the number of parameters and computations.
> >
> > ---
> >
> > > Q8. In experiments, maybe add a CLAPS w/o negative to see which is more important.
> >
> > - We **have included the results of CLAPS w/o negative examples** in the ablation study in **Table 1 and 2 of the revision** .
> >
> > ---
> >
> > > Q10. Why not train as how you generate Table 3 example? This will then better solve the train-test mismatch (exposure bias), although maybe at a cost of slow training.
> >
> > - For preliminary experiments, we generate sentences with adversarial perturbation and maximize $\log (1-p_\theta(y|x))$ as [2], but doing so did not improve any performance.
> >
> > ---
> >
> > > Q11. Some human evaluation on a larger set of generated examples would help. For example, how many hard negatives are actually being recognized as negative by human.
> >
> > - As suggested, we **have conducted a human evaluation of the 20 summaries and 20 questions**generated by our CLAPS and T5-MLE trained for text summarization and QG task.  Specifically, 20 human judges perform blind quality
> > assessment of two sentences generated by the two models, that are presented in a random order. For text summarization, **70\%**of the human annotators chose the sentences generated by our model as better than the baseline, and for QG, **85\%**favored the sentences generated by our model over that of the baseline.
> >
> > - We have also performed **human evaluation of the negative examples (imposters)** to verify the generated ones are  true "negative" example. We randomly sampled 100 negative sentences from our model and  showed them with the corresponding 5 sentences to 20 human annotators. **98%**of the human subjects responded that the generated sentences are grammatically wrong or irrelevant to the input sentences.
> >
> > ---
> >
> > Reference
> >
> >
> > [1] Aghajanyan, Armen, et al. "Better fine-tuning by reducing representational collapse." arXiv preprint arXiv:2008.03156 (2020).
> >
> > [2] Welleck, Sean, et al. "Neural Text Generation With Unlikelihood Training." International Conference on Learning Representations. 2019.

---

> ### Author Response · Authors · 2020-11-24
> **Summary of Response**
>
> This is a summary of what we have provided in response to your feedback
> - We have clarified how our proposed contrastive learning with generating positive and negative examples mitigates the exposure bias problem.
> - We have toned down our claim for exposure bias.
> - We have added discussion about VAT and compare ours against its variants R3F.
> - We have performed additional experiments with the randomly initialized Transformer and report the results.
> - We have conducted human evaluation to qualitatively validate the improvement of our proposed method.
>
>
> We have done our best to address every comment in the initial feedback. The end of the author discussion period is approaching, so if there are any further questions or concerns we would like to do our best to address them as soon as possible.
>
> Thank you for your feedback

---

### Official Review · AnonReviewer1 · 2020-10-29
**This paper presents a method for conditional text generation tasks that aims to over the "exposure bias" problem through contrastive learning where negative examples are generated by adding small perturbations to the input sequence to minimize its conditional likelihood, and positive examples are generated by adding  large perturbations while enforcing it to have a high conditional likelihood.**

**Rating:** 6
**Confidence:** 3

**Review:**

This paper presents a method for conditional text generation tasks that aims to over the "exposure bias" problem through contrastive learning where negative examples are generated by adding small perturbations to the input sequence to minimize its conditional likelihood, and positive examples are generated by adding  large perturbations while enforcing it to have a high conditional likelihood. Experimental results on machine translation, text summarization and question generation show the effectiveness of the proposed approach.

My only concern is that compare to MLE, the improvements either on Table 1 or on Table 2 are relative small. The study in the paper by Massimo Caccia, Lucas Caccia, William Fedus, Hugo Larochelle, Joelle Pineau, Laurent Charlin, Language GANs Falling Short, ICLR 2020 shows that the "exposure bias" problem for text generation by MLE appears to be less of an issue, and simple "temperature sweep" in the softmax significantly boosts the performance and gives pretty good results that beat all language GANs. So I think in the experiments, all results should be compared using the trick of "temperature sweep". Moreover, if diversity is an issue, the results should be compared in the quality-diversity space as did in Language GANs Falling Short paper. Hopefully the authors can address my concern in the rebuttal period.

---

> ### Author Response · Authors · 2020-11-16
> **Response to R1**
>
> We sincerely appreciate your constructive comments. We will reflect your comments in the revision to improve the clarity of the paper. Please refer to the below for detailed discussions
>
> ---
>
> Q1. My only concern is that compare to MLE, the improvements either on Table 1 or on Table 2 are relative small.
>
> - Please note that our **direct baseline is T5-MLE**, which is a pretrained language model trained on a large text corpora, and not Transformer which does not use any pretraining. Transformer performance here is only given as a reference. Please note that our direct competitors are **T5-SSMBA** and **T5-WordDropout Contrastive**, which are **existing contrastive learning methods**for seq2seq learning, which obtain marginal performance improvements over the base T5-MLE. Thus improving the performance by more than 1 point consistently across three tasks here without any additional training data is actually an **impressive**improvement.
>
> - As for the statistical significance, p-value is less than 0.01 for all the experiments.
>
> ---
>
> Q2.The study in the paper by Massimo Caccia, Lucas Caccia, William Fedus, Hugo Larochelle, Joelle Pineau, Laurent Charlin, Language GANs Falling Short, ICLR 2020 shows that the "exposure bias" problem for text generation by MLE appears to be less of an issue, and simple "temperature sweep" in the softmax significantly boosts the performance and gives pretty good results that beat all language GANs. So I think in the experiments, all results should be compared using the trick of "temperature sweep".
>
> - Thanks for pointing out the reference. We have performed the experiments with the **suggested temperature sweep** in the softmax (Please see the **tables below**).  For text summarization, temperature sweep makes some improvements in ROUGE scores, but for others it yields **marginal**performance gains.  Although diversity is another important aspect of language generation, our main focus is on improving the quality of the generated sentences. Compared to the unconditional language generation where the generative model should generate diverse and quality sequences, we believe generating a target sentence that is grounded on the input sentence is more important in conditional language generation.
>
> ---
> NMT
>
> |       | BLEU-1 | BLEU-2 | BLEU-3 | BLEU-4 | Scare-BELU |
> |-------|--------|--------|--------|--------|------------|
> | α=1.0 | 57.76  | 44.45  | 35.12  | 28.21  | 32.43      |
> | α=0.7 | 57.63  | 44.23  | 33.84  | 27.90  | 32.14      |
> | α=2.0 | 56.03  | 42.59  | 33.29  | 26.45  | 30.72      |
> | Ours  | **58.98**  | **45.72**  | **36.39**  | **29.41**  | **33.96**      |
>
> ---
> Question Generation
>
> |       | BLEU-1 | BLEU-2 | BLEU-3 | BLEU-4 | Scare-BELU |
> |-------|--------|--------|--------|--------|------------|
> | α=1.0 | 41.26  | 30.30  | 23.38  | 18.54  | 21.00      |
> | α=0.7 | 40.82  | 29.79  | 22.84  | 17.99  | 20.50      |
> | α=2.0 | 37.35  | 27.20  | 20.79  | 16.36  | 18.41      |
> | Ours  | **42.33**  | **31.29**  | **24.22**  | **19.19**  | **21.55**      |
>
> ---
> Text Summarization
>
> |       | Rouge-1 | Rouge-2 | Rouge-L | Meteor |
> |-------|---------|---------|---------|--------|
> | α=1.0 | 36.10   | 14.72   | 29.16   | 15.78  |
> | α=0.7 | 36.68   | 15.10   | 29.72   | 15.78  |
> | α=2.0 | 34.18   | 13.53   | 27.35   | 14.51  |
> | Ours  | **37.89**   | **15.78**   | **30.59**   | **16.38**  |
>
> ---
>
> References
>
> [1] Ng, Nathan, Kyunghyun Cho, and Marzyeh Ghassemi. "Ssmba: Self-supervised manifold based data augmentation for improving out-of-domain robustness." EMNLP. 2020.
>
> [2] Yang, Zonghan, et al. "Reducing Word Omission Errors in Neural Machine Translation: A Contrastive Learning Approach." ACL. 2019.

---

> > ### Comment · AnonReviewer1 · 2020-11-24
> > **Official Blind Review #1**
> >
> > I think the rebuttal addresses my concern. Thank you.

---

> > > ### Author Response · Authors · 2020-11-25
> > > **Thank you**
> > >
> > > We hope that we have satisfactorily addressed all your concerns. Please let us know there is anything else we need to clarify or provide. We thank you again for your helpful suggestions which significantly improved the quality of our paper.

---

> ### Author Response · Authors · 2020-11-24
> **Summary of Response**
>
> This is a summary of what we have provided in response to your feedback
> - We have added experimental results of another baseline model with temperature sweep.
> - We have performed the T-tests for all results and show all the improvements are statistically significant.
>
> We have done our best to address every comment in the initial feedback. The end of the author discussion period is approaching, so if there are any further questions or concerns we would like to do our best to address them as soon as possible.
>
> Thank you for your feedback

---

### Official Review · AnonReviewer2 · 2020-11-01
**Interesting idea to generate "hard" positive and negative samples for contrastive learning**

**Rating:** 4
**Confidence:** 3

**Review:**

- Overall comments

This paper propose a principled method to generate "hard" positive and negative samples based on conditional likelihood for contrastive learning of seq2seq models, and it shows significant improvements in training conditional text generation tasks compared to naïve approach with random negative samples. Overall, the idea is interesting, and the experiments are well-conducted. However, I still have some detailed questions regarding to the method and experiment as follows:

- Methods:

(1) I am a bit confused with Eq(2). What is $\bf{M}$? Do you mean $x_i$ is the source sentence, $y_i$ is the corresponding target sentence? Is it meaningful to "match" the hidden representation between source and target sentence especially for tasks such as summarization?
Also training with Eq(2) did not involve any decoding process, nor supervising how to decode a sentence. Some form of MLE training (also noted in Eq (9)) seems to be unavoidable which in some sense still relies on teacher forcing..

(2) The proposed method to create positive/negative examples is related to virtual adversarial training (VAT) in NLP:
 *Miyato, Takeru, Andrew M. Dai, and Ian Goodfellow. "Adversarial training methods for semi-supervised text classification." arXiv preprint arXiv:1605.07725 (2016).*
It would be nice to include for discussion or comparison.

(3) For Sec 3.3 & 3.4:
(a) How do we know the perturbed hidden states $\bf{H}$ still lay in the manifold of valid sentences? It is possible the hidden states may not be corresponded to any sentences.
(b) Using the conditional likelihood over the original target sentence to measure the negative samples may also be misleading. For example, it is also possible to get a very different sentence with the same semantic meanings with the target sentence.
(c) What is $\hat{y}$ and $\bar{y}$ in Eq (6) and (7)? Are they different target sentence? Where are they from as the proposed methods did not seem to include decoding.

- Experiments

(1) It seems that all experiments are initialized with T5. Does it mean that the proposed method only works with large scale pre-training? It would be more important to show results with training from scratch.
(2) The results on WMT16 RO-EN do not seem to be too low especially with T5 pre-training which makes the improvement difficult to tell.
(3) For many tasks, the improvements of the proposed method are actually marginal. It may improve the paper by include discussion of statistical significance.
(4) There are also methods such as Reinforcement learning which also aims to overcome the problem of teacher forcing. It should be also discussed in experiments.

---

> ### Author Response · Authors · 2020-11-16
> **Response to R2 (2/3)**
>
> ---
>
> > Q5. Using the conditional likelihood over the original target sentence to measure the negative samples may also be misleading. For example, it is also possible to get a very different sentence with the same semantic meanings with the target sentence.
>
> - Since our model is also trained to maximize the conditional log likelihood, the model assign high probability to sentences that are highly relevant to the input sentence $\mathbf{x}$. So if the loss of a sequence is high (i.e., the conditional log likelihood is low),  then the sequence is not likely target sentence  for the given sentence $\mathbf{x}$. Moreover, perplexity (exponential of negative log likelihood) is used to measure the performance of language model. Therefore, we believe measuring the negativeness with loss is reasonable
>
> ---
>
> > Q6. What are $\hat{y}$ and $\bar{y}$  in Eq (6) and (7)? Are they different target sentences? Where are they from as the proposed methods did not seem to include decoding.
>
> - $\overline{y}$ is a typo of $\hat{y}$. We have corrected them. In Eq.(6) and (7), $\hat{y}$ in $p_\theta(\cdot|\mathbf{x})$ also denotes the random variables  which follows a categorical  distribution $p_\theta(\hat{\mathbf{y}}|\mathbf{x})$ after the perturbation. In order to prevent the perturbed distribution from deviating too much from the original conditional distribution $p_\theta(\mathbf{y}| \mathbf{x})$, we minimize the KL divergence between them as described in Eq.(7). Also, as this is an end-to-end model, learning of the latent embedding space does affect the decoder.
>
> > Q7. It seems that all experiments are initialized with T5. Does it mean that the proposed method only works with large scale pre-training? It would be more important to show results with training from scratch.
>
> - Since finetuning the pretrained language models becomes de facto standard for various tasks of natural language processing, including classification (natural language understanding) and generation, we focus on the pretrained language model. Moreover, several works [4],[5] requires pretrained language models for their method.
>
> - However, to show that our method can be used with models trained from scratch, we have performed the experiments with a **randomly initialized Transformer**with exactly the same architecture as T5. Please see the result of **scratch-T5-MLE** and **scratch-CLAPS** from general response. Our method consistently improves the performance of randomly initialized network.
>
> ---
>
> > Q8. The results on WMT16 RO-EN do not seem to be too low especially with T5 pre-training which makes the improvement difficult to tell.
>
> - Although the performance of pretrained T5 model is high, our proposed method further improves BLEU scores by more than 1 point in **all measures (BLEU-n)**.
>
> ---
>
> > Q9. For many tasks, the improvements of the proposed method are actually marginal. It may improve the paper by include
> discussion of statistical significance.
>
> - Please note that our direct baseline is **T5-MLE, which is a pretrained language model**trained on a large text corpora, and **not Transformer**which does not use any pretraining. Making consistent improvements of more than 1 point over a pretrained language model on three tasks in all measures, is actually impressive, especially since the baseline T5 model is already very strong. The existing contrastive learning or data augmentation baselines for seq2seq learning, including  contrastive baseline with word-dropout [7] (T5-WordDropout Contrastive) and  ssmba [6] (T5-SSMBA), yield marginal improvements (0.01 to 0.1), or even underperforms the base model (T5-MLE), while our method significantly improves the performance in all settings we considers, in all measures.
> - We outperform another strong baseline **R3F [5]**, proposed in a concurrent ICLR 2021 submission (https://openreview.net/forum?id=OQ08SN70M1V) and outperforms existing methods that leverage adversarial perturbations such as SMART and FreeLB.
> - For statistical significance, we performed the T-tests for all results, and all improvements are **statistically significant, with the p-value of less than 0.01**.

---

> > ### Author Response · Authors · 2020-11-17
> > **Response to R2 (3/3)**
> >
> > > Q10. There are also methods such as Reinforcement learning which also aims to overcome the problem of teacher forcing. It should be also discussed in experiments.
> >
> > - We have **included the RL-based methods [1] for overcoming the exposure bias**in the related work section. For preliminary experiments, we train the seq2seq model with [1], but the improvement is very marginal over the base T5-model. The followings are results for text summarization with this RL-based approach. We would like to emphasize again that the improvements obtained using our model are significant, in comparison to all methods and baselines we consider, as it is very difficult to increase the score even by 1 point over the base T5.
> >
> > Text Summarization
> >
> > |       | Rouge-1 | Rouge-2 | Rouge-L | Meteor |
> > |-------|---------|---------|---------|--------|
> > | RL[1]     | 36.26   | 14.71   | 29.16   | 15.60  |
> > | MLE  |  36.10    |14.72    | 29.16   | 15.78  |
> > | Ours  | **37.89**  | **15.78**   | **30.59**   | **16.38**  |
> >
> >
> > ---
> >
> > References
> >
> > [1] Paulus, Romain, Caiming Xiong, and Richard Socher. "A Deep Reinforced Model for Abstractive Summarization." International Conference on Learning Representations. 2018.
> >
> > [2] Yu, Lantao, et al. "Seqgan: Sequence generative adversarial nets with policy gradient." Thirty-first AAAI conference on artificial intelligence. 2017.
> >
> > [3] Guo, Jiaxian, et al. "Long Text Generation via Adversarial Training with Leaked Information." AAAI. 2018.
> >
> > [4]  Miyato, Takeru, Andrew M. Dai, and Ian Goodfellow. "Adversarial training methods for semi-supervised text classification." arXiv preprint arXiv:1605.07725 (2016).
> >
> >
> > [5] Aghajanyan, Armen, et al. "Better fine-tuning by reducing representational collapse." arXiv preprint arXiv:2008.03156 (2020).
> >
> > [6] Ng, Nathan, Kyunghyun Cho, and Marzyeh Ghassemi. "Ssmba: Self-supervised manifold based data augmentation for improving out-of-domain robustness." EMNLP. 2020.
> >
> > [7] Yang, Zonghan, et al. "Reducing Word Omission Errors in Neural Machine Translation: A Contrastive Learning Approach." ACL. 2019.
> >
> > [8] Jiang, Haoming, et al. "Smart: Robust and efficient fine-tuning for pre-trained natural language models through principled regularized optimization." arXiv preprint arXiv:1911.03437 (2019).
> >
> > [9] Zhu, Chen, et al. "Freelb: Enhanced adversarial training for natural language understanding." International Conference on Learning Representations. 2019.

---

> ### Author Response · Authors · 2020-11-17
> **Response to R2 (1/3)**
>
> We sincerely appreciate your constructive comments. We respond to the individual comments below:
>
> ---
>
> > Q1. I am a bit confused with Eq(2). What is M? Do you mean $x_i$ is the source sentence and  $y_i$  is the corresponding target sentence? Is it meaningful to "match" the hidden representation between source and target sentence especially for tasks such as summarization?
>
> - As described in the paper, $x^{(i)}$ and $y^{(i)}$ denote the input and target text sequences (sentences). $\mathbf{M}$ is an output of encoder $f$ given the input sentence $x^{(i)}$ and thus is the hidden representation of the source sentence $x^{(i)}$.
>
> ---
>
> > Q2. Also training with Eq(2) did not involve any decoding process, nor supervising how to decode a sentence.
>
> - The constastive learning of the **latent space**of the encoder-decoder model **will affect the generated text sequences**since for conditional text generation, the generated texts **depends on the latent embedding of the input**text sequence. We provide a way of better learning the latent embedding space, such that we can map latent embedding of a semantically similar output text sequence to be close to the latent embedding of the input text sequence. Moreover, the modification of the latent space will affect the decoder since our model is an end-to-end encoder-decoder architecture.
>
> ---
>
> > Q3. Some form of MLE training (also noted in Eq (9)) seems to be unavoidable which in some sense still relies on teacher forcing.
>
> - Our work aims to **mitigate**the problems introduced with teacher forcing, not to completely replace the teacher forcing. We have made this clear in the revision. Existing works  [1], [2], [3] which tackle the exposure bias problem also still rely on teacher forcing. However, contrary to teacher forcing where the model is trained with only ground truth tokens, with our proposed method, the seq2seq model is exposed to various hidden representations of both positive and negative examples automatically generated using our method, and thus can better learn the latent semantic space for text generation.
>
> ---
>
> > Q3. The proposed method to create positive/negative examples is related to virtual adversarial training (VAT) in NLP: Miyato, Takeru, Andrew M. Dai, and Ian Goodfellow. "Adversarial training methods for semi-supervised text classification." It would be nice to include for discussion or comparison.
>
> - The goal of our work is completely different from VAT [4], although how we add the perturbation may look similar. VAT [4] leverage adversarial examples as augmented data and **enforce consistency** across the predictions of original **unlabeled data** and its perturbation.
> - CLAPS (ours), on the other hand, aims to **distinguish between**the original examples and the adversarial examples with small perturbations, thus using the adversarial sample as a negative example to better guide the text generation. To differentiate our negative examples from conventional adversarial examples, we call them as **imposters**.
>
> - Since VAT [4],  SMART[8], FreeLB[9] are targeting text classification,  they are incomparable to ours. Instead, we implement the method "R3F" [5] that is similar to but outperform them (Please see Table 2 in https://arxiv.org/abs/2008.03156) and target sequence generation.
>
> - We report the performance or our CLAPS in comparison to R3F in the "Additional Experimental Results (https://openreview.net/forum?id=Wga_hrCa3P3&noteId=N4Lq73J_MU-)".
>
> ---
>
> > Q4. How do we know the perturbed hidden states $H$ still lay in the manifold of valid sentences? It is possible the hidden states may not be corresponded to any sentences.
>
> - Table 3 shows the examples of the generated sentences (denoted as Imp.) from the perturbed embedding with greedy decoding. We can observe that distant targets generate valid sentences, while imposters sometimes generate grammatically incorrect sentences. This is beneficial in improving the quality of the conditionally generated texts, since using such grammatically incorrect sentences as negative examples can guide the model to generate grammatically correct sentences.

---

> ### Author Response · Authors · 2020-11-24
> **Summary of Response**
>
> Summary of response for R2
> This is a summary of what we have provided in response to your feedback
> - We have clarified how our proposed contrastive learning with generating positive and negative examples could mitigate the exposure bias problem.
> - We have added discussion about VAT and compare ours to its variant R3F.
> - We have performed the T-tests for all results and show all the improvements are statistically significant.
> - We have added experimental results for randomly initialized Transformer.
> - We have corrected some typos.
>
> We have done our best to address every comment in the initial feedback. The end of the author discussion period is approaching, so if there are any further questions or concerns we would like to do our best to address them as soon as possible.
>
> Thank you for your feedback

---

> ### Author Response · Authors · 2020-11-25
> **The end of the discussion phase approaching.**
>
> Dear reviewer
>
> We sincerely appreciate your efforts in reviewing our paper, and your constructive comments. We have responded to your comments and faithfully reflected them in the revision, and provided additional experimental results that you have requested.
> Could you please go over our responses, new results in the common response, and the revision, since we have less than 11 hours left to have interactive discussions?  Please let us know there is anything else we need to clarify or provide.
>
> Thanks, authors

---

### Author Response · Authors · 2020-11-17
**Discussion of VAT, SMART, FreeLB & Additional Experimental Results**

We provide experimental results with additional baselines suggested by the reviewers.

**Comparison to Virtual Adversarial Training (VAT) [1], SMART [3], FreeLB [4], and R3F [5]**

- VAT utilizes the adversarial examples as augmented data, and enforce consistency across the predictions across original unlabeled example and its perturbation, for semi-supervised learning of a **text classification**model.

- SMART [3] and FreeLB [4] leverage adversarial examples to induce smoothness of classifier for**text classification**, and **R3F [5] (a concurrent submission to ICLR 2021, https://openreview.net/forum?id=OQ08SN70M1V) improves upon them by using Gaussian random noise (that does not require PGD) and considering seq2seq learning**. Since R3F [2] largely outperforms SMART and FreeLB (Please see Table 2 in https://arxiv.org/abs/2008.03156) we have performed experiments against it, in the Tables below. Our method largely outperforms R3F, which is a strong baseline that outperforms SMART and FreeLB, on all tasks.

- Ours is different from the above methods in that we use the slightly perturbed samples as **negative examples** and thus we train the model to differentiate texts embedded closely with different semantics instead of being robust/invariant to them. That is why we name them as **imposters** to differentiate them from conventional adversarial examples. The samples we maximize the similarity to, instead, are the samples far away from the original samples with similar semantics. Thus our work is a method to automatically generates positive/negative samples for constrastive learning of seq2seq model for the model to generate high-quality texts, rather than obtaining a model robust to small perturbations.

---

**Comparison to Caccia et al. 2020 [5]**
- [5] empirically show that MLE trained language models with fine-tuned temperatures (α) for softmax outperform GAN-based language models specifically proposed to tackle the exposure bias problem. To show that our method is superior over it, we have compared against MLE models with varying temperatures in the tables below.

- We have also performed experiments with a randomly initialized Transformer network to show that our method consistently improves the performance of base model, whether they are pretrained or not.

---

The tables below shows the experiments for all baselines we previously described, which shows the clear superiority of our CLAPS over all existing baselines. We have included the results in the Experiments section of the revision (Table 1 and 2).

NMT

|       | BLEU-1 | BLEU-2 | BLEU-3 | BLEU-4 | Scare-BELU |
|-------|--------|--------|--------|--------|------------|
| scratch-T5-MLE| 51.62  | 37.22  | 27.76  | 21.13  | 25.34      |
| scratch-CLAPS         | 53.42  | 39.57  | 30.24  | 23.59  | 27.61      |
| T-MLE(α=1.0) [5]| 57.76  | 44.45  | 35.12  | 28.21  | 32.43      |
| T5-MLE(α=0.7) [5]| 57.63  | 44.23  | 33.84  | 27.90  | 32.14      |
| T5-MLE(α=2.0) [5]| 56.03  | 42.59  | 33.29  | 26.45  | 30.72      |
| R3F [2]  | 58.07  | 44.86  | 35.37  | 28.66  | 32.99      |
| Ours  | **58.98**  | **45.72**  | **36.39**  | **29.41**  | **33.96**      |

---
Question Generation

|       | BLEU-1 | BLEU-2 | BLEU-3 | BLEU-4 | Scare-BELU |
|-------|--------|--------|--------|--------|------------|
|T-MLE(α=1.0) [5] | 41.26  | 30.30  | 23.38  | 18.54  | 21.00      |
|T5-MLE(α=0.7) [5]| 40.82  | 29.79  | 22.84  | 17.99  | 20.50      |
|T5-MLE(α=2.0) [5]| 37.35  | 27.20  | 20.79  | 16.36  | 18.41      |
|R3F [2] | 41.00 | 30.15| 23.26|18.44|20.97|
| Ours  | **42.33**  | **31.29**  | **24.22**  | **19.19**  | **21.55**      |

---
Text Summarization

|       | Rouge-1 | Rouge-2 | Rouge-L | Meteor |
|-------|---------|---------|---------|--------|
| scratch-T5-MLE  | 31.44  | 11.07  | 25.18 | 13.01 |
| scratch-CLAPS         | 33.52  | 12.59  | 26.91 |14.18  |
| T5-MLE(α=1.0) [5] | 36.10   | 14.72   | 29.16   | 15.78  |
| T5-MLE(α=0.7) [5] | 36.68   | 15.10   | 29.72   | 15.78  |
| T5-MLE(α=2.0) [5] | 34.18   | 13.53   | 27.35   | 14.51  |
|R3F [2]      | 36.96   |15.12    |   29.76 | 15.68  |
| Ours  | **37.89**   | **15.78**   | **30.59**   | **16.38**  |

---

References
[1] Miyato, Takeru, Andrew M. Dai, and Ian Goodfellow. "Adversarial training methods for semi-supervised text classification." arXiv preprint arXiv:1605.07725 (2016).

[2] Aghajanyan, Armen, et al. "Better fine-tuning by reducing representational collapse." arXiv preprint arXiv:2008.03156 (2020).

[3] Jiang, Haoming, et al. "Smart: Robust and efficient fine-tuning for pre-trained natural language models through principled regularized optimization." arXiv preprint arXiv:1911.03437 (2019).

[4] Zhu, Chen, et al. "Freelb: Enhanced adversarial training for natural language understanding." International Conference on Learning Representations. 2019.

[5] Caccia, Massimo, et al. "Language GANs Falling Short." International Conference on Learning Representations. 2019.

---

### Author Response · Authors · 2020-11-18
**Summary of the Revision**

We really appreciate all the reviewers for their constructive comments. We have responded to the common comments as well as individual comments from the reviewers below, and believe that we have successfully responded most of them. We have included the discussions and results of the suggested experiments in the revision. Here we briefly summarize the updates we have made to the revised version of the paper:

- We  have included the discussions on the difference of our method from **virtual adversarial training**[1] in section 2 and 3.3.

- We have included results of **R3F** (https://openreview.net/forum?id=OQ08SN70M1V) proposed in a concurrent ICLR 2021 submission (Table 1 and 2 of the revision), which is considerably more efficient and outperforms SMART and FreeLB suggested by the reviewers.

- We  have corrected the typo in Equation 7.

- We  have included **four additional baselines** [2], [3], suggested by (R1), (R2) and (R4),  and **results of the state-of-the-art models**in Section 4.2.

- We have included  experimental results of our CLAPS on the  **randomly initialized  T5 models**in Section 4.2, following the suggestion of (R2) and (R4).

- We have included the **human evaluation results** for text summarization and question generation tasks in Section 4.3, as suggested by (R2) and (R4).

- We have included **more examples of the generated texts**in the Appendix, as suggested by (R4).

---

References

[1] Miyato, Takeru, Andrew M. Dai, and Ian Goodfellow. "Adversarial training methods for semi-supervised text classification." arXiv preprint arXiv:1605.07725 (2016).

[2]  Aghajanyan, Armen, et al. "Better fine-tuning by reducing representational collapse." arXiv preprint arXiv:2008.03156 (2020).

[3] Caccia, Massimo, et al. "Language GANs Falling Short." International Conference on Learning Representations. 2019.

---

### Author Response · Authors · 2020-11-23
**The end of the discussion phase approaching.**

Dear Reviewers,

Could you please go over our responses and the revision since we can have interactions with you only by this Tuesday (24th)? We have responded to your comments and faithfully reflected them in the revision, and provided additional experimental results that you have requested. We sincerely thank you for your time and efforts in reviewing our paper, and your insightful and constructive comments.

Thanks, Authors

---

### Author Response · Authors · 2020-11-24
**Less than 24 hours left until the end of the interactive discussion phase**

We sincerely appreciate your efforts in reviewing our paper, and your constructive comments. Could you please go over our responses, new results in the common response, and the revision, since we have less than 24 hours left to have interactive discussions? Could you please let us know if there are any other concerns that we should address? We would be pleased to clarify them and revise our paper by the response deadline.

Thanks, authors

---

### Decision · Program_Chairs · 2021-01-07
**Final Decision**

**Decision:**

Accept (Poster)

**Comment:**

This paper proposes a new method for conditional text generation that uses contrastive learning to mitigate the exposure bias problem in order to improve the performance. Specifically, negative examples are generated by adding small perturbations to the input sequence to minimize its conditional likelihood, while positive examples are generated by adding large perturbations while enforcing it to have a high conditional likelihood.

This paper receives 2 reject and 2 accept recommendations, which is a borderline case. The reviewers have raised many useful questions during the review process, while the authors has also done a good job during the rebuttal to address the concerns. After checking the paper and all the discussions, the AC feels that all the major concerns have been solved, such as more clarification in the paper, more results on non-pretrained models, and small-scale human evaluation.

On one hand, reviewers found that the proposed method is interesting and novel to a certain extent, the paper is also well written. On the other hand, even after adding all the additional results, the reviewers still feel it is not super-clear that results would extend to better models, as most of the experiments are conducted on T5-small, and the final reported numbers in the paper are far from SOTA.

As shown in Table 1 & 2, the AC agrees that the final results are far from SOTA, and the authors should probably also study the incorporation of CLAPS into stronger backbones. On the other hand, the AC also thinks that T5 is already a relatively strong baseline to start with (though it is T5-small), and it may not be necessary to chase SOTA. Under a fair comparison, the AC thinks that the authors have done a good job at demonstrating its improvements over T5-MLE baselines.

As a summary, the AC thinks that the authors have done a good job during the rebuttal. On balance, the AC is happy to recommend acceptance of the paper. The authors should add more careful discussions to reflect the reviewers' comments when preparing the camera ready.